# CHIMERA: STATE SPACE MODELS BEYOND SEQUENCES

## ABSTRACT

Powerful deep learning methods based on Transformers are used to model diverse data modalities such as sequences, images, and graphs. These methods typically use off-the-shelf modules like self-attention, which are domain-agnostic and treat data as an unordered set of elements. To improve performance, researchers employ inductive biases—such as position embeddings in sequences and images, and random walks in graphs—to inject the domain structure, or *topology* into the model. However, these inductive biases are carefully engineered heuristics that must be designed for each modality, requiring significant research effort. In this work, we propose *Chimera*, a unified framework that mathematically generalizes state space models to incorporate the topological structure of data in a principled way. We demonstrate that our method achieves state-of-the-art performance across domains including language, vision, and graphs. Chimera outperforms BERT on the GLUE benchmark by 0.7 points, surpasses ViT by 2.6% on ImageNet-1k classification accuracy, and outperforms all baselines on the Long Range Graph Benchmark with a 12% improvement on PascalVOC. This validates Chimera's methodological improvement which allows it to directly capture the underlying topology, providing a strong inductive bias across modalities. Furthermore, being topologically aware enables our method to achieve a linear time complexity for sequences and images, in contrast to the quadratic complexity of attention.

## 1 INTRODUCTION

Real-world data is heterogeneous, ranging from sequential language data to high-dimensional image data, and structured data of proteins and molecules. Despite this heterogeneity, many domains exhibit an inherent *topology* that encodes the neighborhood of each element (node) of the data. For instance, language and audio have a directed line graph topology, where each node (token) is arranged sequentially (Fig 1a). Similarly, images possess an undirected grid-graph topology, where each node (image patch) is connected to its immediate local neighbors in a grid (Fig 1b). Structured data like proteins have predefined nodes (atoms) and edges (bonds), which constitute their topology (Fig 1c).

Typical approaches to model data build upon Transformers (Vaswani et al., 2017) with self-attention at their core (Devlin et al., 2019; Dosovitskiy et al., 2021; Rampášek et al., 2022). However, since self-attention is permutation invariant, it treats data as an unordered set of elements and completely disregards the data's topology. To address this, significant research effort has focused on developing domain-specific heuristics, such as position embeddings (Su et al., 2023; Devlin et al., 2019), and random walks (Behrouz and Hashemi, 2024; Wang et al., 2024), to serve as the inductive bias for the underlying topology. However, developing these heuristics requires navigating a large search space for each domain. For instance, RoPE embeddings (Su et al., 2023) work well in language (Touvron et al., 2023); in vision, absolute position embeddings are widely used (Dosovitskiy et al., 2021; Heo et al., 2024); while laplacian embeddings are used in graphs (Rampášek et al., 2022). Moreover, given the lack of systematic underpinnings, it is unclear how effectively they capture the underlying topology.

In this paper, we consider the following problem: *"Can we develop a principled method that captures the underlying data topology, and achieves state-of-the-art performance across domains?"*. We propose *Chimera*, a domain-agnostic framework built on recent State Space Models (SSMs)—Mamba-2 (Dao and Gu, 2024a), RetNet (Sun et al., 2023), Linear Attention (LA) (Katharopoulos et al., 2020)—that mathematically generalizes SSMs to *any* topology and achieves state-of-the-art performance across diverse domains including language, images, and graphs. These consistently superior results validate Chimera's methodological improvement which allows it to directly capture the underlying topology, providing a strong inductive bias across various modalities. This contrasts with existing approaches that instead apply attention or SSMs as a black box to "flattened data", supplemented by heuristics.

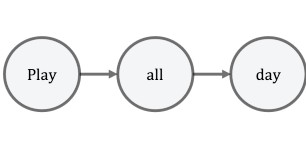 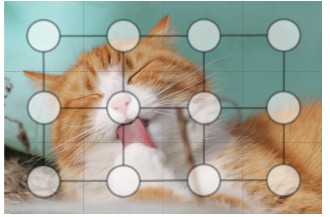 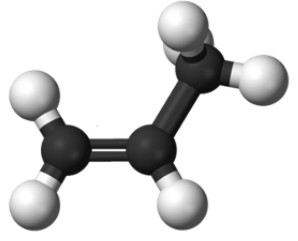

(a) Language (Line Graph)  (b) Images (Grid Graph)  (c) Molecules (General Graph)

Figure 1: Real-world data exhibits inherent topology: (a) language follows a directed line graph, (b) images a grid graph, and (c) structured data like molecules have explicit graph topology.

Furthermore, being topologically aware allows Chimera to leverage the simpler topology of line and grid graphs to avoid "unnecessary computation", thus reducing its computational cost to linear in the number of nodes. This recovers the linear complexity of SSMs while maintaining strong performance.

To derive Chimera, we consider SSMs for causal language modeling and formally show that SSMs *inherently capture the underlying directed line graph topology* through their recurrence structure (Sec 3.2). For this, we leverage the Structured Masked Attention (SMA) representation (Dao and Gu, 2024a): Multiple methods including Mamba-2, RetNet, LA are SSMs, and these SSMs are equivalent to the matrix $\mathbf{M} = \mathbf{L} \odot (\mathbf{Q}\mathbf{K}^T)$ acting on the input, where $\mathbf{Q}$ and $\mathbf{K}$ are the query and key matrices respectively, and $\mathbf{L}$ is a (data dependent) mask matrix. This mask matrix $\mathbf{L}$ is analogous to the causal masked attention matrix used in Transformers. We show that for SSMs, the mask matrix $\mathbf{L}$ can be represented as the resolvent of the adjacency matrix, $\mathbf{A}$, of a directed line graph, i.e., $\mathbf{L} = (\mathbf{I} - \mathbf{A})^{-1} = \sum \mathbf{A}^i$, where $\mathbf{I}$ is the identity matrix. Thus, $\mathbf{L}$ characterizes a specific SSM model and is also equivalent to the resolvent of the adjacency matrix, connecting SSMs and the underlying topology.

We extend this result to generalize SSMs to any topology. Specifically, we appropriately parameterize the adjacency matrix $\mathbf{A}$, and compute the SMA matrix $\mathbf{M} = \mathbf{L} \odot (\mathbf{Q}\mathbf{K}^T)$, where $\mathbf{L} = (\mathbf{I} - \mathbf{A})^{-1}$. Intuitively, $\mathbf{A}_{ij}$ captures the influence between neighbor $i$ and $j$, and the resolvent then accumulates the influence between each pair of nodes through all possible paths between them, thus capturing the underlying topology. We present a detailed scheme to parameterize $\mathbf{A}$ and introduce a normalization scheme for $\mathbf{A}$, which is crucial for the stable training of our method.

Central to Chimera is the computation of the mask matrix whose naive implementation incurs cubic cost. To avoid this, Chimera leverages structure in the topology to significantly speed up this calculation. Specifically, for the class of directed acyclic graphs (DAGs), the resolvent operation can be computed in linear time. This is especially useful for topologies like undirected line graphs and grid graphs, which can be canonically decomposed into multiple DAGs: An undirected line graph decomposes into two directed line graphs (Fig 4), while a grid graph divides into four directed grid graphs (Fig 5). This allows us to implement Chimera in linear time—recovering the complexity of SSMs —while preserving the underlying topology. We further show that for general graphs, we can efficiently compute the finite sum approximation of the resolvent, capturing the global topological structure while achieving performance competitive with state-of-the-art baselines. Overall, we make the following contributions:

- We propose Chimera, a unified framework that generalizes SSMs to any data topology.
- We introduce a technique that leverages the underlying data topology using DAGs to improve the efficiency of Chimera, achieving linear time complexity for sequences and images.
- We validate that Chimera consistently achieves state-of-the-art results across diverse domains including language, images, and graphs: It outperforms BERT (Devlin et al., 2019) by a GLUE score (Wang et al., 2019) of 0.7, surpasses ViT (Dosovitskiy et al., 2021) on ImageNet-1k (Deng et al., 2009) classification by 2.6%, and outperforms strong baselines on the Long Range Graph Benchmark (LRGB) (Dwivedi et al., 2022), notably increasing PascalVOC's F1 score by 12% .

## 2 PRELIMINARIES

In this section, we introduce State Space Models (SSMs), which are recurrent models designed to process sequential data, such as language and audio. We first formulate SSMs in their recurrent form and then introduce the Structured Masked Attention (SMA) (Dao and Gu, 2024a) representation that interprets this recurrence as a matrix $\mathbf{M}$ acting on the input $\mathbf{X}$. In the subsequent section, we use the SMA representation to show that SSMs inherently operate on a directed line graph topology.

## 2.1 OVERVIEW OF STATE SPACE MODELS

SSMs, such as Mamba-2 (Dao and Gu, 2024a), Linear Attention (LA) (Katharopoulos et al., 2020), RetNet (Sun et al., 2023), are recurrent sequence-to-sequence models that feature a linear hidden-state transition function. This linearity enables a hardware-efficient, vectorized implementation of SSMs, allowing them to scale effectively. Furthermore, this transition function is typically data-dependent which is known to improve model performance (Hwang et al., 2024).

Formally, let $X \in \mathbb{R}^{T \times D}$ denote the input sequence of $T$ tokens, where each token has $D$ channels. Let the size of the hidden state be $d$. Let $Y \in \mathbb{R}^{T \times D}$ be the output of the sequence-to-sequence model. Then, SSMs begin by computing the following matrices:

$$B = f_B(X) \in \mathbb{R}^{T \times d}, \quad C = f_C(X) \in \mathbb{R}^{T \times d}, \quad V = f_V(X) \in \mathbb{R}^{T \times d}, \tag{1}$$

where $f_B, f_C, f_V$ are model specific data dependent functions. For instance, in Mamba-2 each of these functions is a composition of a linear projection of $X$ along the channel dimension, followed by a short convolution layer along the sequence dimension and a Swish activation function (Ramachandran et al., 2017). In Dao and Gu (2024a), it was shown that we can view the $B$, $C$, and $V$ matrices as analogs of the key, query, and value matrices in self-attention.

Let $v^i \in \mathbb{R}^T$ denote the input corresponding to channel $i$ (i.e., $v^i = V[:,i]$). For any time $t$, define $B_t = B[t,:]$, $C_t = C[t,:]$, $y_t^i = Y[t,i]$ and $v_t^i = v^i[t]$. Then, the model computes a recurrence, which is a function from $B, C, \Delta, V$ to the output $Y$, starting with the hidden state $h_{-1}^i = 0 \in \mathbb{R}^d$ as,

$$h_t^i = a_t h_{t-1}^i + b_t B_t v_t^i, \tag{2}$$

$$y_t^i = C_t^T h_t^i, \tag{3}$$

where $a_t, b_t$ are model-specific parameters that characterize the SSM. For instance, Linear Attention sets $a_t = b_t = 1$, RetNet chooses $a_t = \gamma$, $b_t = 1$ for some learnable parameter $\gamma$. In contrast, Mamba-2 sets $a_t, b_t$ in a data-dependent manner to implicitly encode a gated memory mechanism known as *selectivity* or the *selection mechanism*. This mechanism allows the model to select and propagate important tokens across long sequences. Specifically, define,

$$\Delta = f_\Delta(X) \in \mathbb{R}^T, \quad a_t = \exp(-\Delta_t) \in \mathbb{R}, \quad b_t = \Delta_t \in \mathbb{R}, \tag{4}$$

where $\Delta$ is the selectivity matrix, and $f_\Delta$ like $f_B, f_C, f_V$ is a data-dependent function. The selection mechanism operates as follows: for an important token, $\Delta_t$ is large, and the model gives more weight to token $t$ while reducing the contribution of the previous hidden state. Conversely, for an unimportant token, $\Delta_t$ is small and the model retains most of the past hidden state, with minimal contribution from token $t$. This allows Mamba-2 to retain important tokens through long recurrences.

## 2.2 SSM IN THE STRUCTURED MASKED ATTENTION REPRESENTATION

In Dao and Gu (2024a), the authors introduced the Structured Masked Attention (SMA) representation, which computes the same function as the SSM recurrence (Eq. 3) described in the previous section but instead interprets the function computation as a matrix $M$ acting on the value matrix $V$.[1] They demonstrate that such an $M$ is a function of $B, C, \Delta$ matrices (defined above) and can be expressed as $M = L \circ CB^T$, where $L$ is a data-dependent mask matrix derived from the $\Delta$ matrix.

Formally, define $\bar{B}_t = b_t B_t$, and recall from Section 2.1 that $b_t = \Delta_t$, $a_t = \exp(-\Delta_t)$ for Mamba-2; $b_t = 1$, $a_t = \gamma$ for RetNet; and $b_t = 1$, $a_t = 1$ for Linear Attention. Then the output $Y$ computed by the recurrence (Eq. 3) can be vectorized as,

$$Y = MV = (L \odot C\bar{B}^T)V, \tag{5}$$

where the structured mask matrix $L_{ij} = \mathbf{1}[i \geq j] \Pi_{j < k \leq i} a_k$, for all $i, j$,

$$L = \begin{bmatrix} 1 & 0 & 0 & \cdots & 0 \\ a_1 & 1 & 0 & \cdots & 0 \\ a_1 a_2 & a_2 & 1 & \cdots & 0 \\ \vdots & \vdots & \vdots & \ddots & \vdots \\ a_1 a_2 \cdots a_{T-1} & a_2 a_3 \cdots a_{T-1} & a_3 a_4 \cdots a_{T-1} & \cdots & 1 \end{bmatrix}. \tag{6}$$

The SMA representation is useful because, as we will demonstrate in Section 3, it neatly isolates the effect of the underlying topology within the recurrence computation into the mask matrix $L$. This property allows us to generalize SSMs to arbitrary topologies by appropriately formulating the mask $L$.

---

[1]Note that not all SSMs have an SMA representation, but you focus throughout this paper on ones that do (LA, RetNet, Mamba-2) and use we will use "SSMs" to refer specifically to this restricted class.

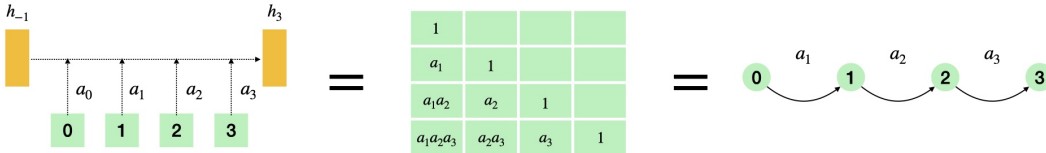

Figure 2: SSMs inherently operate on a directed line graph: SSMs modeling a sequence of tokens (left), the structured mask matrix (center), Chimera on a directed line graph (right)

## 3  CHIMERA: BUILDING MODELS FOR ANY TOPOLOGY

In this section, we introduce Chimera, a unified framework that generalizes SSMs to any arbitrary topology, enabling the development of performant models across diverse domains. Existing approaches such as Behrouz and Hashemi (2024); Devlin et al. (2019); Liu et al. (2021), treat attention and SSMs as black-box modules operating on fixed topologies such as sets or sequences and rely on heuristics to incorporate structural information. In contrast Chimera opens up this black box and mathematically adapts it to handle any topology.

To motivate Chimera, we first analyze the setting of SSMs applied to the causal language modeling task. We show that the recurrence in SSMs naturally operates on a directed line graph topology. To formalize this result, we first define the resolvent of a linear operator and interpret its action when this operator is a weighted adjacency matrix of a topology.

### 3.1  RESOLVENT OF AN ADJACENCY MATRIX ACCUMULATES INFLUENCE

A graph topology consists of a set of nodes $\mathcal{V}$ that represent data elements, and edges $\mathcal{E}$ that encode the underlying topological structure. We conceptualize the associated adjacency matrix $\boldsymbol{A} \in \mathbb{R}^{|\mathcal{V}| \times |\mathcal{V}|}$ as *capturing the influence between neighboring nodes*. Specifically, $\boldsymbol{A}_{ij}$ is the influence that node $j$ has on node $i$, for each edge $(i,j)$. The natural desideratum then is to extend the notion of influence to all node pairs by incorporating the graph's structure, accounting for all possible paths between them. To model this cumulative influence, we introduce the concept of the resolvent of a linear operator

**Definition 3.1** (Resolvent of a Linear Operator (Reed and Simon, 1980)). *Let $\boldsymbol{A} \in \mathbb{R}^{T \times T}$ be a linear operator, $\boldsymbol{I}$ the identity operator, and $\lambda$ a complex number. Then, the resolvent operator is defined as:*
$$R(\lambda, \boldsymbol{A}) = (\lambda \boldsymbol{I} - \boldsymbol{A})^{-1}, \tag{7}$$
*which exists for all complex numbers $\lambda$ that are not in the spectrum of $\boldsymbol{A}$, i.e., $\lambda \notin \sigma(\boldsymbol{A})$. In this work, we set $\lambda = 1$ to remain in the field of real numbers, and this is done without loss of generality, as any choice of $\lambda$ is equivalent upto scaling of the model.*

We now demonstrate how the resolvent operator captures the influence between any two nodes in the graph. Observe that the resolvent operation can be expanded using the Liouville-Neumann series if the operator norm of the adjacency matrix, $\|\boldsymbol{A}\|$, is less than 1,
$$R(1, \boldsymbol{A}) = (\boldsymbol{I} - \boldsymbol{A})^{-1} = \sum_{k=0}^{\infty} \boldsymbol{A}^k. \tag{8}$$

Intuitively, each term $\boldsymbol{A}^k$ in this expansion represents the influence between any two nodes $i$ and $j$ through all paths of length exactly $k$ connecting them. This is formalized in Proposition 3.2.

**Proposition 3.2** ($\boldsymbol{A}^k$ accumulate influence through paths of length $k$). *Given the weighted adjacency matrix $\boldsymbol{A} \in \mathbb{R}^{T \times T}$ of a graph $\mathcal{G} = (\mathcal{V}, \mathcal{E})$ with $|\mathcal{V}| = T$, the $(i,j)^{th}$ entry of $\boldsymbol{A}^k$ is:*
$$(\boldsymbol{A}^k)_{ij} = \sum_{p_1, p_2, \ldots, p_{k-1}} \boldsymbol{A}_{i p_1} \boldsymbol{A}_{p_1 p_2} \cdots \boldsymbol{A}_{p_{k-1} j},$$
*where $(p_1, \ldots, p_{k-1})$ is an ordered sequence of vertices forming a path of length $k$ from node $i$ to $j$.*

Therefore, the series $(\boldsymbol{I} - \boldsymbol{A})^{-1}$ (Eq. 8) sums up the influence of node $i$ on node $j$ over all possible paths and path lengths. Additionally, we also note that Eq. 8 provides a sufficient condition for the existence of the resolvent: the series converges when the operator norm of $\boldsymbol{A}$ is less than one.

### 3.2  SSMs OPERATE ON A DIRECTED LINE GRAPH

We now show that SSMs naturally operate on a directed line graph. Specifically, let $\mathcal{V}$ be the set of tokens, and $\mathcal{E}$ be the edges connecting token $t$ to the next token $t+1$. The weighted adjacency matrix is defined as $\boldsymbol{A}_{s,t} = \mathbf{1}_{[t=s+1]} a_t$, where $a_t$ is the method-specific parameter described in Section 2.2.

We recall from Section 2.2 that SSMs can be represented as the SMA matrix $\mathbf{M} = \mathbf{L} \odot (\mathbf{CB}^T)$. We make the key observation that $\boldsymbol{L}$ *is precisely the resolvent of* $\boldsymbol{A}$, that is $\boldsymbol{L} = (\boldsymbol{I} - \boldsymbol{A})^{-1}$. This ties SSMs to the directed line graph topology, with the mask matrix encoding the topology (Fig 2).

**Proposition 3.3.** *Under the notation established in Section 2, consider a weighted directed graph $\mathcal{G}$ with nodes $\mathcal{V} = \{0, \cdots, T-1\}$, edges $\mathcal{E} = \{(i-1, i)|i \in \mathcal{V}, i > 0\}$, and the edge weights $\mathcal{W} = \{w_{i-1 \rightarrow i} = a_i | i \in \mathcal{V}, i > 0\}$. Let $\boldsymbol{A}$ be the weighted adjacency matrix of incoming edges,*

$$\boldsymbol{A} = \begin{bmatrix} 0 & 0 & 0 & \cdots & 0 \\ a_1 & 0 & 0 & \cdots & 0 \\ 0 & a_2 & 0 & \cdots & 0 \\ \vdots & \vdots & \vdots & \ddots & \vdots \\ 0 \cdots 0 & 0 \cdots 0 & 0 & a_{T-1} & 0 \end{bmatrix}, \tag{9}$$

*then $\boldsymbol{L} = \sum_{i=0}^{\infty} \boldsymbol{A}^i = (\boldsymbol{I} - \boldsymbol{A})^{-1}$, and consequently, $\boldsymbol{y} = ((\boldsymbol{I} - \boldsymbol{A})^{-1} \odot \boldsymbol{C}\bar{\boldsymbol{B}}^T)\boldsymbol{V}$.*

We can interpret this result intuitively: in a directed line graph, there is exactly one path between tokens $i, j$ with $i < j$, and the corresponding entry in $\boldsymbol{L}$, $\boldsymbol{L}_{ij} = \prod_{i \geq k > j} a_k$, reflects the cumulative influence of the intervening tokens along this path. Furthermore, $\boldsymbol{L}_{ij} = 0$ for $i < j$ restricts influence in the forward direction, ensuring that the model remains causal. This shows that SSMs inherently operate on a directed line graph with the $\boldsymbol{L}$ matrix encoding the topology.

## 3.3 GENERALIZING SSMS TO ARBITRARY TOPOLOGIES

We now build on Proposition 3.3 to generalize SSMs to arbitrary topologies. Specifically, we compute the resolvent of an "appropriately parameterized" adjacency matrix, $\boldsymbol{A}$, and model the output in the SMA representation as $((\boldsymbol{I} - \boldsymbol{A})^{-1} \odot (\boldsymbol{C}\bar{\boldsymbol{B}}^T))\boldsymbol{V}$. In this section, we focus on the parameterization of $\boldsymbol{A}$ for arbitrary topologies and ensuring the numerical stability of the method, particularly in cases of non-invertibility or poor conditioning of $\boldsymbol{I} - \boldsymbol{A}$.

Formally, consider a graph $\mathcal{G} = (\mathcal{V}, \mathcal{E})$ with $|\mathcal{V}| = T$ nodes, where each node has $D$ channels. Let $d$ denote the generalized hidden state size. For each node, we compute the following matrices,

$$\boldsymbol{B} = f_B(\boldsymbol{X}) \in \mathbb{R}^{T \times d}, \ \boldsymbol{C} = f_C(\boldsymbol{X}) \in \mathbb{R}^{T \times d}, \ \boldsymbol{V} = f_V(\boldsymbol{X}) \in \mathbb{R}^{T \times d}, \ \Delta = f_\Delta(\boldsymbol{X}) \in \mathbb{R}^T, \tag{10}$$

where the functions $f_B$, $f_C$, $f_V(\boldsymbol{X})$, $f_\Delta$ are linear projections applied to the input, followed by a local graph convolution over neighboring nodes and a Swish activation as chosen in Mamba-2. Our parameterization is inspired by Mamba-2 (Dao and Gu, 2024a)—one of the latest iterations of SSMs—as it features selectivity, which allows it to effectively model long-range dependencies. However, we note that our approach can generalize any SSM with an SMA representation.

We parameterize the $\boldsymbol{A}$ matrix for each edge $(i, j) \in \mathcal{E}$ as,

$$\boldsymbol{A}_{ij} = \frac{\exp(-\Delta_i) + \exp(-\Delta_j)}{2}, \tag{11}$$

to incorporate context from both ends of the edge $(i, j)$. To add directionality to the edge representation and to further increase the representational power of our model, we can also maintain two (different) $\Delta$'s such that $\boldsymbol{A}_{ij} = (\exp(-\Delta_i^{(1)}) + \exp(-\Delta_j^{(2)}))/2$.

Note that the matrix $\boldsymbol{I} - \boldsymbol{A}$ may be either non-invertible or poorly conditioned, which could hinder the stable training of the model. To address this, we introduce a data-dependent normalization parameter $\Psi = f_\Psi(\boldsymbol{X}) \in \mathbb{R}^T$, computed similarly to $\Delta$, and perform a row-wise normalization of the adjacency matrix using $\Psi$. Specifically, for each row $i \in [T]$, we apply:

$$\boldsymbol{A}[i, :] = \frac{\gamma \boldsymbol{A}[i, :]}{\mathbf{1}^T \boldsymbol{A}[i, :] + \exp(-\Psi_i)},$$

where $\gamma$ is a scaling hyperparameter. In the following proposition, we show that this normalization guarantees the convergence of the Neumann series for the adjacency matrix $\boldsymbol{A}$.

**Proposition 3.4.** *Under Gaussian initialization, the row-wise normalization strategy ensures that $\|\boldsymbol{A}\| < 1$ and $\|(\boldsymbol{I} - \boldsymbol{A})^{-1}\|$ is bounded with probability greater than $1 - \Phi(\frac{-1}{\gamma})$.*

We provide the proof for this proposition in Appendix A.1. Finally, we compute the resolvent matrix $\boldsymbol{L} = (\boldsymbol{I} - \boldsymbol{A})^{-1}$ and the output $\boldsymbol{y}$ as $(\boldsymbol{L} \odot \boldsymbol{C}\bar{\boldsymbol{B}}^T)\boldsymbol{V}$.

## 4 CHIMERA WITH IMPROVED EFFICIENCY

While Chimera works with arbitrary graph topologies, directly computing the resolvent incurs a cubic cost in the number of nodes. However, we show that we can significantly reduce this computational cost when the underlying topology is more structured. Specifically, we consider the class of directed acyclic graphs (DAGs), a generalization of directed line graphs, and show that the resolvent can be computed in linear time, matching the complexity of SSMs like Mamba-2.

### 4.1 CHIMERA ON DAGS

We tailor Chimera to DAGs with a specialized normalization scheme and an algorithm to compute the output in linear time. Our choice of DAGs is motivated by the fact that topologies such as undirected line and grid graphs can be canonically decomposed into DAGs: a line graph divides into two directed line graphs (Fig 4) and a grid graph divides into four directed grid graphs (Fig 5). This decomposition enables Chimera to operate efficiently with a linear complexity while preserving topology.

Formally, consider a DAG $\mathcal{G} = (\mathcal{V}, \mathcal{E})$ with $|\mathcal{V}| = T$ nodes, each with $D$ channels and a hidden state size of $d$. For any node $i$, let $p(i)$ be the set of its parents. Let $B, C, V, \Delta$ be the input projections as defined in Section 3. We define the adjacency matrix $A$ as $A_{ij} = \exp(-\Delta_i[j])$ for each $(i,j) \in \mathcal{E}$, and set $\bar{B}_i = \Delta_i B_i$ for each node $i$. We first show that the resolvent $(I - A)^{-1}$ exists.

**Proposition 4.1.** *For a DAG, $A$ is nilpotent, that is $A^T = 0$. Therefore, the inverse $(I - A)^{-1}$ exists and is given by the finite sum:*

$$L = (I - A)^{-1} = \sum_{t=0}^{T-1} A^t. \tag{12}$$

As in previous sections, we compute the output of the model as $y = (L \odot (C\bar{B}^T))V$. Furthermore, this method admits an equivalent recurrent view (Prop. 4.2).

**Proposition 4.2.** *Our method computes the following recurrence on each channel $v$ of $V$:*

$$h_i = \sum_{j \in p(i)} A_{ij} h_j - \bar{B}_i v_i, \qquad y_i = C_i^T h_i, \tag{13}$$

*where $h_l = 0$ for all leaf nodes $l$.*

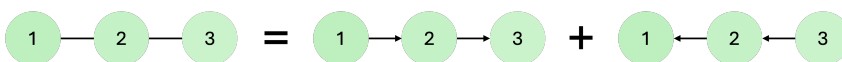

Observe that while the resolvent always exists, its entries can become exceedingly large which can cause numerical instabilities. Recall from Section 3.1 that each $L_{ij}$ represents the cumulative sum of all paths from node $j$ to $i$, and in the worst case, the number of such paths grows exponentially with distance. To address this, we introduce a normalization scheme that is built directly into the recurrence:

Figure 3: Recurrence on DAGs

**Proposition 4.3.** *The normalized method computes the following recurrence:*

$$h_i = \frac{1}{\sqrt{|p(i)|}} \sum_{j \in p(i)} (A_{ij} h_j - \ln(A_{ij}) B_i v_i), \tag{14}$$

$$y_i = C_i^T h_i. \tag{15}$$

*This normalization ensures that $\mathrm{Var}(C_i^T h_i) \leq 1$ under the assumption that the vectors $\{B_i v_i, C_i\}_i$ are i.i.d. Gaussians, that is $B_i v_i, C_i \sim \mathcal{N}(0, I_d)$.*

The proof follows by induction on the time step $t$, where at each time step, we ensure that the output variance is bounded by 1, $\mathrm{Var}(C_i^T h_i) \leq 1$, which guarantees that the output remains a well-behaved random variable. We provide the detailed proof in Appendix A.2. To incorporate this normalization in the SMA representation, we define,

$$\bar{A} = \frac{1}{\sqrt{|p(i)|}} A, \quad \bar{B} = \frac{\ln(A_{ij})}{\sqrt{|p(i)|}} B, \quad L = (I - \bar{A})^{-1}, \tag{16}$$

and compute the output $y = (L \odot (C\bar{B}^T))V$.

Figure 4: The undirected line graph structure (Left). The canonical DAG decomposition (Right)

### 4.1.1 CHIMERA IS EFFICIENT ON DAGS

Finally, we highlight that DAGs are a particularly important case of Chimera because of additional efficiency benefits, both theoretically and through optimized implementations.

**Linear-time Complexity** The intuition for the linear complexity is that the resolvent operation for DAGs is *finite* because of the lack of cycles. From the adjacency matrix perspective, $A$ is nilpotent, i.e. $A^k = 0$, where $k$ is the diameter of the graph (Prop 4.1). Since Chimera can be equivalently viewed as a recurrence on the DAG, the resolvent operation converges after one pass through the graph in the topological order which takes linear time.

**Proposition 4.4.** *The Chimera structured mask matrix $L$ can be computed in $O(|\mathcal{V}| + |\mathcal{E}|)$ complexity where $|\mathcal{V}|, |\mathcal{E}|$ is the number of vertices and edges of the graph, respectively.*

The proof is provided in Appendix A.3. We note that the linear-time complexity of Mamba can be seen as a special case of Theorem 4.4 specialized to the directed line graph, where both $|\mathcal{V}|$ and $|\mathcal{E}|$ is equal to the sequence length.

**Improving Efficiency Through Matrix Multiplications** Finally, we note that on modern hardware accelerators such as GPUs and TPUs, various computational algorithms can have different efficiency tradeoffs. For example, on directed line graphs, the naive computation of SSMs and RNNs as a recurrence is not parallelizable and is inefficient in practice (Gu and Dao, 2023). In the case of DAGs, we present a technique to reduce both the forward and backward pass for Chimera to leverage only matrix multiplications which are heavily optimized on modern accelerators.

**Theorem 4.5.** *In case of Chimera on DAGs, the forward pass can be computed with $O(\log(dia(\mathcal{G})))$ matrix multiplications where $dia(\mathcal{G})$ is the diameter of the graph (i.e. length of the longest path), and the backward pass can be computed with $O(1)$ matrix multiplications.*

**Backward pass.** The local update rule of backpropagation requires applying the chain rule through the matrix inverse operation, in particular, using the following identity applied to $Y = (I - A)$,

$$\frac{\partial Y^{-1}}{\partial \theta} = -Y^{-1} \frac{\partial Y}{\partial \theta} Y^{-1} \tag{17}$$

Because $Y^{-1}$ is already computed in the forward pass, it can be cached, and then the marginal cost of the local backpropagation is simply two extra matrix multiplications.

**Forward pass.** To compute $L = (I - A)^{-1}$ more efficiently for DAGs, we leverage the equivalence of Neumann series to the series $L = I + A + A^2 + \cdots$, which comes to a finite sum for DAGs due to the nilpotence of $A$ matrix. We compute this sum more efficiently using the "squaring trick" as,

$$(I - A)^{-1} = (I + A)(I + A^2)(I + A^4) \cdots (I + A^k), \tag{18}$$

where $k$ is the smallest power of 2 larger than the graph diameter $dia(\mathcal{G})$. This can be computed using $O(\log(dia(\mathcal{G})))$ matrix multiplications to compute the powers of $A$ for powers-of-two exponents, and then $O(\log(dia(\mathcal{G})))$ matrix multiplications to multiply together the right-hand side.

### 4.1.2 APPROXIMATE CHIMERA FOR GENERAL TOPOLOGY

While DAGs allow for efficient computation in structured domains like images and language, directly computing the resolvent $L$ for general graph topology remains computationally expensive. To address this, we use a finite-sum relaxation of the resolvent operator and truncate its corresponding Neumann series sum (Eq. 8) at some maximum power $k \in \mathbb{N} > 0$. Specifically, let $A$ be the adjacency matrix of the graph topology defined in Section 3.3, then,

$$L = \sum_{i=0}^{\infty} A^i \approx \hat{L} = \sum_{i=0}^{k} A^i. \tag{19}$$

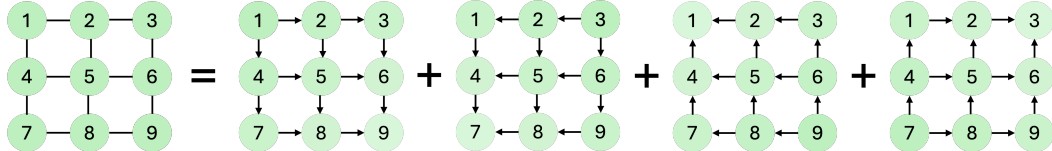

Figure 5: Grid graph (left). The canonical 2D-DAG decomposition of the grid graph (right). These graphs are sufficient to capture the influence between all pairs of nodes in the undirected grid graph.

We choose $k = \text{diam}(\mathcal{G})$, the diameter of the graph, to ensure that $\hat{L}$ has access to the global structure of the graph, that is, it includes contributions from every edge and node in the graph.

**Proposition 4.6.** *If $k \geq dia(\mathcal{G})$, then for any pair of nodes $(i,j)$, if $L_{ij} > 0$ in the original method, then $\hat{L}_{ij} > 0$ in the finite-sum relaxation.*

As in Section 4.1.1, we can compute this approximation efficiently using the squaring trick:
$$\hat{L} = (I + A)(I + A^2)(I + A^4) \cdots (I + A^p), \tag{20}$$
where $p$ is the smallest power of 2 larger than or equal to the graph diameter $\text{dia}(\mathcal{G})$. This reduces the computational cost of the method to $O(\log(\text{dia}(\mathcal{G})))$ matrix multiplications.

## 5 EXPERIMENTS

In this section, we will demonstrate that *directly incorporating topology is a powerful inductive bias for diverse domains* such as language, images and graphs, eliminating the need for domain-specific heuristics. Chimera consistently achieves state-of-the-art performance in these domains. On language, it outperforms BERT on the GLUE benchmark (Wang, 2018) by a GLUE score of 0.7. On images, it surpasses ViT models on the ImageNet-1k classification (Deng et al., 2009) task by 2.6%. On general graphs, Chimera outperforms strong baselines on the Long Range Graph Benchmark (Dwivedi et al., 2021) which highlights our method's ability to model long range interactions on graphs. Notably, our method improves upon PascalVOC dataset's F1 score by over 12%.

### 5.1 MASKED LANGUAGE MODELING

We evaluate Chimera on bidirectional language modeling, which has a line graph topology (Fig. 4). We test two Chimera variants: the general method[2] (Sec. 3) applied to an undirected line graph, and the DAG method (Sec. 4.1), applied to the canonical DAG decomposition of undirected line graphs into two directed line graphs and summing the resolvents of both DAGs (Fig. 4). Both methods are trained on the Masked Language Modeling (MLM) (Devlin et al., 2019) task on the C4 dataset (Raffel et al., 2020) for 70k steps, following the recipe used in M2 (Fu et al., 2023). The models are then fine-tuned on the GLUE benchmark. We refer the reader to Appendix C for details.

Table 1: Comparing Chimera on the undirected line graph (UG), and on DAG decomposed directed line graphs (DAG) with other state-of-the-art models including M2 (Fu et al., 2023), MLP-Mixer (Tolstikhin et al., 2021), FNet (Lee-Thorp et al., 2022), BERT (Devlin et al., 2019) on GLUE benchmark

| Method | #Params | Pretrain | | GLUE Tasks | | | | | | | | GLUE |
|---|---|---|---|---|---|---|---|---|---|---|---|---|
| | | $\mathcal{L}_{ce}$ | Acc (%) | MNLI | QNLI | QQP | RTE | SST2 | MRPC | COLA | STS | Avg |
| BERT-Base | 110M | 1.59 | 67.3 | 84.1 | **89.8** | **91.2** | 77.2 | 91.2 | 87.5 | 54.6 | **88.9** | 83.2 |
| MLP-Mixer | 112M | 1.77 | 63.5 | 77.2 | 82.4 | 87.6 | 67.3 | 90.5 | 86.5 | 43.0 | 85.2 | 77.5 |
| FNet | 112M | 1.94 | 61.3 | 74.9 | 82.1 | 85.7 | 63.6 | 87.6 | 86.4 | 42.7 | 83.1 | 75.8 |
| M2 | 116M | 1.65 | 65.9 | 80.5 | 86.0 | 87.0 | 69.3 | 92.3 | 89.2 | 56.0 | 86.9 | 80.9 |
| Chimera (UG) | 110M | 1.49 | 68.5 | 83.63 | 88.98 | 89.32 | 73 | 93.67 | 89.4 | 56.95 | 88.82 | 82.97 |
| Chimera (DAG) | 110M | 1.46 | **68.9** | **84.11** | 89.78 | 89.77 | **77.98** | **93.69** | **90.36** | **57.08** | 88.68 | **83.93** |

From Table 1, observe that while BERT outperforms other linear baselines such as M2, MLP-Mixer, FNet it does so with an additional quadratic cost. In contrast, Chimera achieves the best of both worlds, incurring a linear time complexity while achieving state-of-the-art performance. This capability arises from two key factors: first, our parameterization of the adjacency matrix allows the model to effectively modulate the influence between tokens in the sequence, leading to strong performance. Second, the structured nature of the adjacency matrix enables a fast, linear-time resolvent operation, improving the method's computational efficiency. Additionally, note that our undirected graph (UG) variant performs competitively with BERT while surpassing other recent baselines with a linear time complexity.

### 5.2 IMAGENET-1K CLASSIFICATION

We evaluate Chimera on the ImageNet-1k (Deng et al., 2009) classification task that has a grid graph topology. We compare Chimera applied to the 2D-DAG decomposition (Figure 5) topology against state-of-the-art ViT based models, specifically we use ViT-B which has 88M parameters. We also compare against other latest linear time baselines like Hyena (Poli et al., 2023), S4 (Gu et al., 2022)

---

[2]We use a slightly modified normalization scheme for the undirected line graph method to allow for larger selectivity values in the adjacency matrix. See Appendix B.1 for details

in Table 2. We note that *all these baselines flatten the image into a 1D sequence and apply 1D sequence models, and do not take into account the underlying topology*. For our experiments, we simply replace the SSD layer in the Mamba block introduced in Dao and Gu (2024a) with Chimera, and use the ViT-B training recipe with no additional hyperparameter tuning.

Table 2: Top-1, Top-5 accuracies of various methods on ImageNet-1K.

| Method (88M) | Top-1 (%) | | Top-5 (%) | |
|---|---|---|---|---|
| | Acc | Acc$_{EMA}$ | Acc | Acc$_{EMA}$ |
| ViT-B | 78.8 | 80.6 | 94.2 | 95.2 |
| S4-ViT-B | 79.4 | 80.4 | 94.2 | 95.1 |
| Hyena-ViT-B | 78.4 | 76.4 | 94.0 | 93.0 |
| Chimera-ViT-B | **81.4** | **82.1** | **95.4** | **95.9** |

Table 3: Ablation: Comparing 2D grid structure with 1D flattening of patches.

| Method (22M) | Top-1 (%) | | Top-5 (%) | |
|---|---|---|---|---|
| | Acc | Acc$_{EMA}$ | Acc | Acc$_{EMA}$ |
| Fwd (1D) | 73.8 | 73.8 | 91.6 | 91.6 |
| Fwd & Rev (1D) | 76.5 | 75.6 | 93.4 | 92.8 |
| 2D DAG | **77.8** | **76.7** | **93.9** | **93.5** |

Table 2 shows that Chimera's 2D-DAG decomposition outperforms ViT by 2.6%. We note that our method does not require any additional position embeddings which are still an active area of research for ViT (Heo et al., 2024). Furthermore, we outperform methods such as Hyena (Poli et al., 2023) by 3%, and S4 (Gu et al., 2022) by 2% that linearize the data and then apply an SSM on it.

To demonstrate the importance of incorporating topology, we perform an ablation where we progressively degrade the grid-graph structure, observing a monotonic drop in performance. We consider three topologies: **2D DAG** is the 2D DAG decomposition that retains the grid structure (Fig 5, right); **Fwd & Rev (1D)** flattens the grid into a 1D sequence with bidirectional edges like ViT (Fig 6, top); **Fwd (1D)** is a 1D graph with only forward edges (Fig 6, bottom). We observe from Table 3 that as the topology is lost, the accuracy drops from 77.8% (2D-DAG) to 76.5% (Fwd & Rev) to 73.8% (Fwd).

### 5.3    Long Range Graph Benchmark

We evaluate Chimera on the Long Range Graph Benchmark (LRGB) (Dwivedi et al., 2022). This benchmark comprises tasks designed to challenge models in their ability to effectively capture both local and long-range interactions within graph structures. We compare against convolution-based (GCN Kipf and Welling (2016), GatedGCN Bresson and Laurent (2017)), Transformer-based (GraphGPS Rampášek et al. (2022)) , Mamba-based (Graph-Mamba Wang et al. (2024), Graph Mamba Behrouz and Hashemi (2024)), and other baselines like GINE Hu et al. (2019), as well as their hyperparameter tuned versions introduced in Tönshoff et al. (2023). These baselines incorporate topology using a variety of techniques: convolution ones use local aggregation, transformer ones use local and global aggregation via position embeddings, and Mamba ones use "data flattening" along with random walks, position embeddings, and local encodings. The diversity of these methods highlights the significant research effort dedicated to heuristics to incorporate topology, in contrast to our unified approach.

We show that Chimera achieves state-of-the-art results across all LRGB tasks (Table 4). Notably, we observe that on tasks such as Peptides-Func and Peptides-Struct, where convolution-based models typically outperform transformers, Chimera outperforms or matches their performance. Furthermore, on tasks like PascalVOC and COCO where transformers do well, Chimera consistently surpasses all baselines, with a more than 12% improvement on PascalVOC. This validates our grounded approach which effectively captures both local and global information.

In Table 5, we evaluate the approximate variant of Chimera with a finite-sum relaxation (Sec 4.1.2) that truncates the Neumann series at the average graph diameter of the graph. We show that the approximation variant matches the strong transformer baseline of GraphGPS, however fully leveraging the entire graph structure in Chimera provides clear performance benefits.

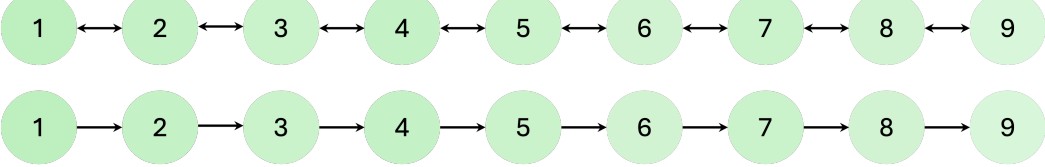

Figure 6: Progressively destroying the 2D grid graph topology. *Fwd & Rev* (top): 1D flattened grid with bidirectional edges. *Fwd* (bottom): 1D flattened grid graph with only forward edges.

Table 4: Evaluation of Chimera on LRGB Tasks (9). The first section shows the best performing numbers cited in the papers that introduce the given baselines. The second section shows the result of better hyperparameter tuned baselines introduced by Tönshoff et al. (30). Finally, we also compare with other baselines that use SSMs as a blackbox replacement for a Transformer.

| Method (< 500k params) | Peptides-Func | Peptides-Struct | PascalVOC-SP | COCO-SP |
| --- | --- | --- | --- | --- |
| | AP ($\uparrow$) | MAE ($\downarrow$) | F1 ($\uparrow$) | F1 ($\uparrow$) |
| GCN (17) | 0.5930±0.0023 | 0.3496±0.0013 | 0.1268±0.0060 | 0.0841±0.0010 |
| GINE (14) | 0.5498±0.0079 | 0.3547±0.0045 | 0.1265±0.0076 | 0.1339±0.0044 |
| Gated-GCN (2) | 0.5864±0.0077 | 0.3420±0.0013 | 0.2873±0.0219 | 0.2641±0.0045 |
| SAN+LapPE (18) | 0.6384±0.0121 | 0.2683±0.0043 | 0.3230±0.0039 | 0.2592±0.0158 |
| Exphormer (26) | 0.6527±0.0043 | 0.2481±0.0007 | 0.3975±0.0037 | 0.3430±0.0108 |
| GPS+BigBird (24) | 0.5854±0.0079 | 0.2842±0.0130 | 0.2762±0.0069 | 0.2622±0.0008 |
| GraphGPS+Transformer (24) | 0.6575±0.0049 | 0.2510±0.0015 | 0.3689±0.0131 | 0.3774±0.0150 |
| GCN (30) | 0.6860±0.0050 | **0.2460±0.0007** | 0.2078±0.0031 | 0.1338±0.0007 |
| Gated-GCN (30) | 0.6765±0.0047 | 0.2477±0.0009 | 0.3880±0.0040 | 0.2922±0.0018 |
| GINE (30) | 0.6621±0.0067 | 0.2473±0.0017 | 0.2718±0.0054 | 0.2125±0.0009 |
| GraphGPS+Transformer (30) | 0.6534±0.0091 | 0.2509±0.0014 | 0.4440±0.0054 | 0.3884±0.0055 |
| Graph-Mamba (35) | 0.6739±0.0087 | 0.2478±0.0016 | 0.4191±0.0126 | 0.3960±0.0175 |
| Graph Mamba (1) | **0.7071±0.0083** | 0.2473±0.0025 | 0.4393±0.0112 | 0.3974±0.0101 |
| Chimera (Ours) | **0.7021±0.003** | **0.2460±0.0002** | **0.496±0.007** | **0.3977±0.016** |

Table 5: Ablation: Chimera with approximate resolvent is competitive with the Transformer baseline.

| Method | Peptides-Func | Peptides-Struct | PascalVOC-SP | COCO-SP |
| --- | --- | --- | --- | --- |
| | AP ($\uparrow$) | MAE ($\downarrow$) | F1 ($\uparrow$) | F1 ($\uparrow$) |
| GraphGPS+Transformer | 0.6534±0.0091 | 0.2509±0.0014 | 0.4440±0.0054 | 0.3884±0.0055 |
| Chimera (Ours) | **0.7021±0.003** | **0.2460±0.0002** | **0.496±0.007** | **0.3977±0.016** |
| Chimera (Approx) | 0.6709±0.0089 | 0.2521±0.0006 | 0.4508±0.0367 | 0.3709±0.0009 |

## 6 CONCLUSION AND FUTURE WORK

In this work, we propose Chimera, a unified framework that mathematically generalizes State Space Models (SSMs) to incorporate the underlying data topology. Unlike previous approaches that rely on carefully engineered heuristics and treat attention and SSMs as black boxes, our method breaks open this black box by providing a principled, domain-agnostic framework for modeling diverse data modalities. We show that Chimera achieves state-of-the-art performance across domains including language, vision, and graph tasks, consistently surpassing highly tuned domain-specific baselines, which validates our premise and the proposed solution. Furthermore, we also show that for structured domains like sequences and images, Chimera has an efficient linear complexity by leveraging our DAG decomposition technique, recovering the complexity of SSMs like Mamba-2.

Our work is the first step toward developing unified models for diverse data modalities. We believe that extending the DAG decomposition technique to general graphs to achieve linear complexity is an exciting direction for future work. Furthermore, we hope that the research community applies Chimera to more domains with an inherent underlying topology, and establishes Chimera as a strong baseline for further research in those domains.

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

# A DEFERRED PROOFS

## A.1 PROOF OF PROPOSITION 3.4

*Proof.* Let $\epsilon_i \sim \mathcal{T}(\mathbf{0}, \mathbf{I}_T)$ be $T$ i.i.d. random Gaussian vectors. Assuming Gaussian initialization for the adjacency matrix $\mathbf{A}$, it can be expressed as:

$$\mathbf{A}[i,:] = \frac{\gamma \epsilon_i}{\|\epsilon_i\| + \exp(-\Psi_i)}. \tag{21}$$

We first show that $\|\mathbf{A}\| \leq \gamma < 1$. From the concentration of the norm of a Gaussian random vector, with high probability $\|\epsilon_i\| \geq \sqrt{T}$ for all tokens $i$. Since $\exp(-\Psi_i) \geq 0$, $\|\epsilon_i\| + \exp(-\Psi_i) \geq \sqrt{T}$. Consider any unit vector $\mathbf{u}$, then

$$\|\mathbf{A}\mathbf{u}\| = \sum_{i=1}^{T} \frac{\gamma \epsilon_i^T \mathbf{u}}{\|\epsilon_i\| + \exp(-\Psi_i)} \leq \gamma \sum_{i=1}^{T} \frac{\epsilon_i}{\sqrt{T}} \leq \gamma \frac{\sqrt{T}\epsilon}{\sqrt{T}} = \gamma\epsilon < 1, \tag{22}$$

with probability greater than $1 - \Phi(\frac{-1}{\gamma})$, were $\epsilon_i, \epsilon \sim \mathcal{N}(0,1)$. Finally, since the operator norm of $\|\mathbf{A}\|$ is less than one, we apply Banach's Lemma to get,

$$\|(\mathbf{I} - \mathbf{A})^{-1}\| \leq \frac{1}{1 - \|\mathbf{A}\|}, \tag{23}$$

which implies that the inverse exists. □

## A.2 PROOF OF PROPOSITION 4.3

*Proof.*

$$\text{Var}(\mathbf{C}_i^T \mathbf{h}_i) = \frac{1}{|p(i)|} \left( \sum_{j \in p(i)} \mathbf{A}_{ij} \text{Var}(\mathbf{C}_i^T \mathbf{h}_j) + \ln(\mathbf{A}_{ij}) \text{Var}(\mathbf{C}_i^T \mathbf{B}_i v_i) \right), \tag{24}$$

$$= \frac{1}{|p(i)|} \left( \sum_{j \in p(i)} \mathbf{A}_{ij} \text{Var}(\mathbf{C}_j^T \mathbf{h}_j) + \frac{2}{d} \ln(\mathbf{A}_{ij}) \right), \tag{25}$$

where we have used the fact that $\text{Var}(\mathbf{C}_j^T \mathbf{h}_j) = \text{Var}(\mathbf{C}_i^T \mathbf{h}_i)$, and that the variance of $\mathcal{X}^2$ distribution with $d$ degrees of freedom is $2d$. Let $d \geq 4$, then

$$\text{Var}(\mathbf{C}_i^T \mathbf{h}_i) \leq \frac{1}{|p(i)|} \left( \sum_{j \in p(i)} \mathbf{A}_{ij} + \frac{2}{d} \ln(\mathbf{A}_{ij}) \right) \leq \frac{1}{|p(i)|} \sum_{j \in p(i)} 1 \leq 1, \tag{26}$$

where we have used the fact that $\mathbf{A}_{ij} \in [0,1]$. □

## A.3 PROOF OF PROPOSITION 4.4

*Proof.* In the structured masked attention (SMA) framework Dao and Gu (2024b), the computational complexity is the cost of the matrix-vector multiplication by the mask matrix $\mathbf{L} = (\mathbf{I} - \mathbf{A})^{-1}$. In the case of DAGs, $\mathbf{A}$ is (up to conjugation by a permutation) a *lower-triangular* matrix with $|\mathcal{E}|$ (number of edges) non-zero entries. It suffices to analyze the cost of computing the multiplication $\mathbf{y} = (\mathbf{I} - \mathbf{A})^{-1}\mathbf{x}$. Rewriting as $(\mathbf{I} - \mathbf{A})\mathbf{y} = \mathbf{x}$, $\mathbf{y}$ can be computed through Gaussian elimination on the matrix $\mathbf{I} - \mathbf{A}$, which takes time proportional to the number of non-zero entries or $|\mathcal{V}| + |\mathcal{E}|$.

In graph terminology, this operation can be viewed as a dynamic programming algorithm to propagate features through the SSM update, where the ordering of edges to perform the update rule is given by the Gaussian elimination ordering. □

# B  ADDITONAL EXPERIMENTS

## B.1  MLM: CHIMERA ON UNDIRECTED LINE GRAPHS

For an undirected line graph (Figure 4, left), the adjacency matrix $\boldsymbol{A}$ takes the following form:

$$\boldsymbol{A} = \begin{bmatrix} 0 & a_{12} & 0 & \cdots & 0 \\ a_{21} & 0 & a_{23} & \cdots & 0 \\ 0 & a_{32} & 0 & \cdots & 0 \\ \vdots & \vdots & \vdots & \ddots & \vdots \\ 0\cdots 0 & 0\cdots 0 & 0 & a_{T-1,T} & 0 \end{bmatrix}.$$

As discussed in Section 3.3, to ensure the existence of $(\boldsymbol{I}-\boldsymbol{A})^{-1}$, we introduced a row-wise sum normalization strategy, wherein we normalized each row of the adjacency matrix with $\sum_j \boldsymbol{A}_{ij} + \Psi_i$. However, since this constraint is designed for general graphs, it is not sufficiently expressive. Therefore, we instead use a strictyly more expressive constraint for line graphs which enforces $\boldsymbol{A}_{ij} \cdot \boldsymbol{A}_{ji} + \Psi_i \leq \frac{1}{4}$ on each simple cycle of the graph.

**Proposition B.1.** *Under the above constraint, the inverse $(\boldsymbol{I}-\boldsymbol{A})^{-1}$ exists as for any two nodes, the sum of all paths between them is upper bounded by $\sum_i (1/4)^i \leq 1/3$.*

## B.2  IMAGENET: PARAMETER SHARING ABLATION

We study the trade-off between sharing parameters for $\boldsymbol{B}, \boldsymbol{C}$ across different graphs as a domain-dependent design choice. We explore four settings: *No sharing, Complete sharing, Row-wise sharing, and Diagonal sharing* across the four DAGs. From Table 6, we observe that diagonal sharing achieves the best performance, indicating it strikes the optimal tradeoff between parameter sharing and other modes of increasing expressivity for modeling image data.

| Method (22M) | Top-1 (%) | | Top-5 (%) | |
|---|---|---|---|---|
| | Acc | Acc$_{EMA}$ | Acc | Acc$_{EMA}$ |
| None | 77.10 | 76.13 | 93.55 | 93.15 |
| Complete | 77.25 | 76.09 | 93.75 | 93.21 |
| Row-wise | 77.46 | 76.57 | 93.76 | 93.37 |
| Diagonal | **77.80** | **76.69** | **93.87** | **93.53** |

Table 6: Ablation: Diagonal parameter sharing works best.

## C ARCHITECTURAL DETAILS

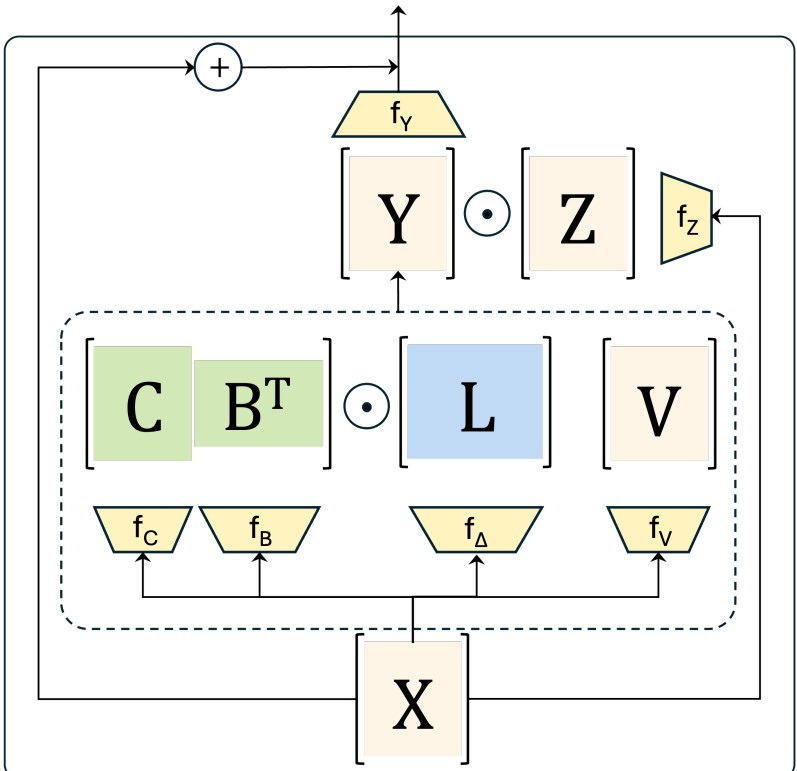

Figure 7: Chimera's Architecture: The output of the Chimera layer is embedded within the gated block introduced in Mamba-2 (Dao and Gu, 2024a). Here $X$ matrix denotes the input to the block, and $f_c, f_B, f_\Delta$ and $f_V$ are data dependent projections defined in Section 2. The operator $\odot$ denotes element-wise multiplications between matrices, and $\oplus$ defines addition. The output from the Chimera layer is passed through a Gated-MLP, a final projection $f_Y$, followed by a residual connection.

### C.1 MASKED LANGUAGE MODELING

Table 7: Architectural and Training Details for BERT-B and Chimera on MLM

| Parameter | BERT-B (110M) | Chimera (110M) |
|---|---|---|
| Model dimension ($d_{\text{model}}$) | 768 | 768 |
| Layers | 12 | 23 |
| Max sequence length | 128 | 128 |
| Num Heads | 12 | 12 |
| Head size | 64 | 64 |
| Optimizer | Decoupled AdamW | Decoupled AdamW |
| Learning rate | $5e{-}4$ | $8e{-}4$ |
| Optimizer momentum | $\beta_1{=}0.9, \beta_2{=}0.98$ | $\beta_1{=}0.9, \beta_2{=}0.98$ |
| Weight decay | $1e{-}5$ | $1e{-}5$ |
| Batch size | 4096 | 4096 |
| Learning rate schedule | Linear decay with warmup | Linear decay with warmup |
| Training steps | 70k | 70k |
| MLM Probability | 0.3 | 0.3 |

In Table 7, we provide the architectural and training details for BERT-B and Chimera on the MLM task. For both the models, we follow the M2 recipe from Fu et al. (2023), adjusting the number of layers to

12 for BERT-B and 23 for Chimera to control for the number of parameters. We conducted a small sweep to fine-tune the learning rate for Chimera, choosing $8e-4$ over BERT-B's $5e-4$.

## C.2 IMAGENET-1K CLASSIFICATION

For the image classification experiments, we largely follow the ViT-B recipe with the following adjustments as shown in Table 8: To control for the number of parameters, we adjust the number of layers from 12 for ViT-B to 22 for Chimera. Additionally, we reduce the Cutmix augmentation from 1.0 to 0.1, as Chimera's stronger inductive bias mitigates the risk of overfitting.

In Table 9, we present the reduced setting used for our ablation studies in Tables 6 and 3, where we match the number of parameters of ViT-S (22M).

Table 8: Hyperparameters used for ViT-B and Chimera for ImageNet-1k classification task

| Parameter | ViT-B (88M) | Chimera (88M) |
|---|---|---|
| Image size | $224^2$ | $224^2$ |
| Optimizer | AdamW | AdamW |
| Optimizer momentum | $\beta_1, \beta_2 = 0.9, 0.999$ | $\beta_1, \beta_2 = 0.9, 0.999$ |
| Weight init | trunc. normal (std=0.02) | trunc. normal (std=0.02) |
| Learning rate | $1e-3$ | $1e-3$ |
| Weight decay | 0.05 | 0.05 |
| Batch size | 1024 | 1024 |
| Training epochs | 310 | 310 |
| Learning rate schedule | cosine decay | cosine decay |
| Warmup epochs | 10 | 10 |
| Warmup schedule | linear | linear |
| Patch Size | 16 | 16 |
| Layers | 12 | 22 |
| Num Heads | 12 | 12 |
| Droppath | 0.3 | 0.3 |
| Randaugment | (9,0.5,layers=2) | (9,0.5,layers=2) |
| Mixup | 0.8 | 0.8 |
| Cutmix | 1.0 | 0.1 |
| Random erasing | 0.25 | 0.25 |
| Label smoothing | 0.1 | 0.25 |
| Stochastic depth | 0.1 | 0.25 |
| Exp. mov. avg (EMA) | 0.99996 | 0.99996 |

Table 9: Key differences between the original and the ablation setting for Chimera

| Parameter | Chimera-S (2D) |
|---|---|
| Model dimension ($d_{\text{model}}$) | 384 |
| Number of layers | 22 |
| Number of Heads | 3 |
| Droppath | 0.1 |

## C.3 LONG RANGE GRAPH BENCHMARK

To train Chimera on the Long Range Graph Benchmark we follow a similar training recipe to that provided in Rampášek et al. (2022) where we replace the Transformer layers with Chimera layers. Moreover, in line with the baselines, we make sure that our models have less than $500k$ parameters. While training Chimera on graphs we remove the Gated-MLP layer $Z$ defined in Figure 7. We did this to keep our training recipe as close to that provided in Rampášek et al. (2022) and highlight the effectiveness of Chimera. The hyperparameters used to train Chimera are provided in Table 10.

Table 10: Hyperparameters running Chimera on the Long Range Graph Benchmark

|  | **Peptides-Func** | **Peptides-Struct** | **PascalVOC-SP** | **COCO-SP** |
|---|---|---|---|---|
| Learning Rate | 0.001 | 0.001 | 0.001 | 0.001 |
| Optimizer | Adam | Adam | Adam | Adam |
| dropout | 0.1 | 0.1 | 0.1 | 0.1 |
| #layers | 2 | 2 | 4 | 4 |
| hidden dim. | 256 | 256 | 128 | 128 |
| head depth | 2 | 2 | 2 | 2 |
| batch size | 32 | 32 | 32 | 32 |
| #epochs | 250 | 250 | 200 | 200 |
| norm | BatchNorm | BatchNorm | BatchNorm | BatchNorm |
| MPNN | GCN | GCN | GCN | GCN |
| #Param. | 461k | 447k | 498k | 498k |

