## 4. Is the optimization landscape of Transformers (ViT/BERT) better than that of SSMs (Chimera). Could the authors add a figure on training loss to compare them?

In our experiments, we did not observe any deterioration in the optimization landscapes of Chimera. We have attached the training loss curves and the eval metrics curves for BERT-B (110M) and Chimera (DAG, 110M) trained on the C4 dataset below (Table 1 in the paper)

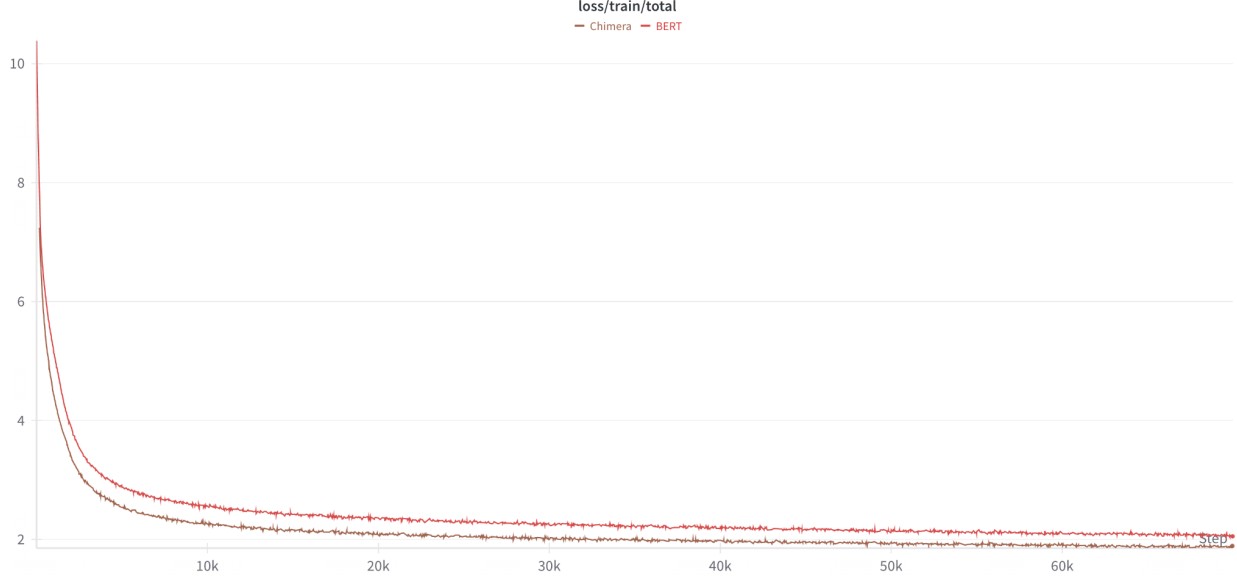

Figure 1: Train Loss for Chimera (Brown) and BERT (Red)

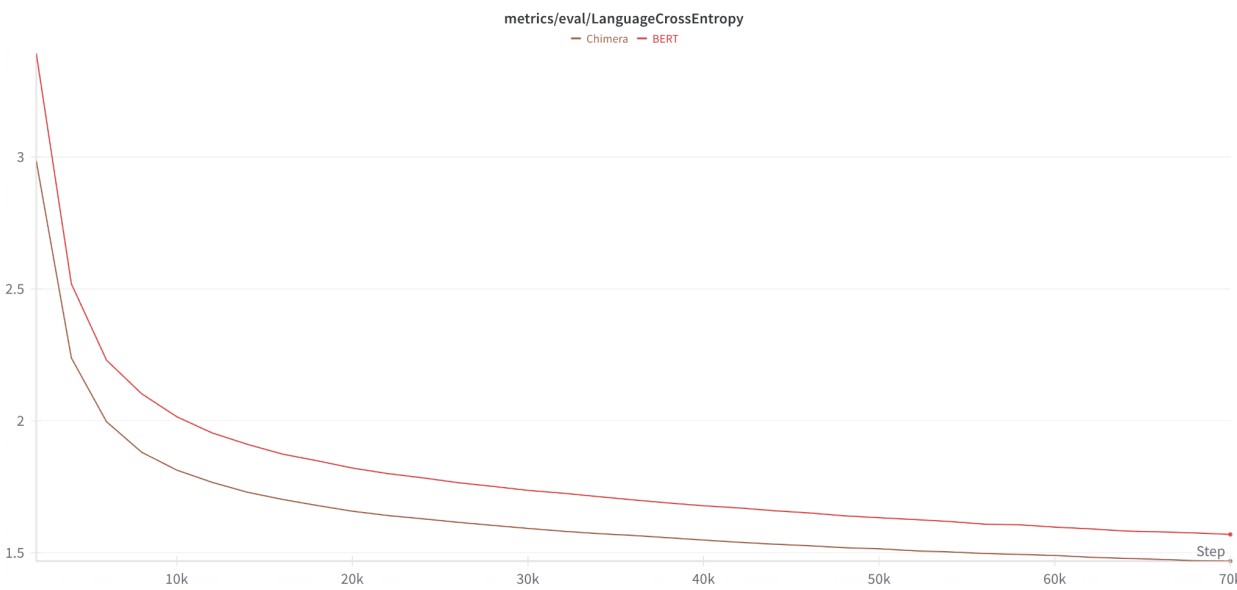

Figure 2: Cross Entropy Loss on the Eval Dataset for Chimera (Brown) and BERT (Red)

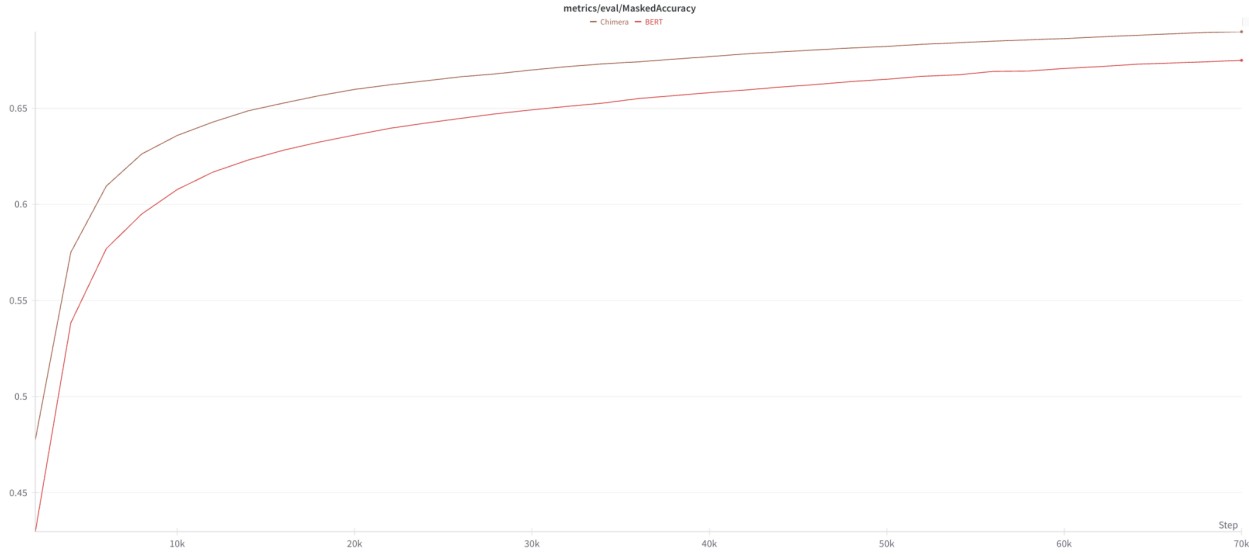

Figure 3: Masked accuracy on the Eval Dataset for Chimera (Brown) and BERT (Red)