# OpenReview forum: "Chimera: State Space Models Beyond Sequences"
_ICLR.cc/2025/Conference — Submitted to ICLR 2025_

### Official Review · Reviewer_uXiv · 2024-10-28

**Soundness:** 3
**Presentation:** 3
**Contribution:** 3
**Rating:** 6
**Confidence:** 3

**Summary:**

Chimera is a new framework that generalizes state space models to effectively incorporate the topology of diverse data types, such as sequences, images, and graphs, into its structure. Unlike traditional methods that treat data as an unordered set, Chimera captures the inherent topology of data, leading to methodological improvements and state-of-the-art performance across various domains. It achieves this by using a structured masked attention mechanism that parameterizes the adjacency matrix to accumulate influences between nodes, allowing for linear time complexity and outperforming existing models like BERT and ViT in benchmarks.

**Strengths:**

This paper introduces a versatile, fundamental model architecture that stands in contrast to the commonly utilized Transformer models. The effectiveness of this architecture is demonstrated through experiments in vision, graph analysis, and natural language processing. This work represents a significant contribution to the ML community. Addressing the minor weaknesses below would strongly support its acceptance.

**Weaknesses:**

1. The fairness of the configurations in Tables 7 and 8 appears to be questionable. (Question 1)

2. The scalability of the proposed architectures needs to be addressed. (Questions 2, 3, and 4)

**Questions:**

1. In the experiments, the Chimera model possesses approximately twice the depth of the ViT model. It is a well-established fact in CV and NLP that depth contributes more to model power than width. While the comparison between SSMs and Transformers in terms of depth is common, it prompts the question: would the Chimera model maintain its superior performance if it had the same number of layers and parameters as the Transformer models?

2. Memory and computational efficiency are often cited advantages of SSMs over Transformers. The inclusion of both theoretical and empirical comparisons between the two would enhance the contribution of this paper.

3. Previous works involving SSMs and the Mamba model have indicated that larger models do not always outperform smaller ones, possibly due to various factors. Could the authors conduct experiments with smaller models in CV and NLP to investigate this phenomenon?

4. Some previous works show that the optimization landscape of Transformers is better than that of SSMs. Could the authors add a figure on training loss to compare the convergence of ViT/BERT with Chimera?

---

> ### Author Response · Authors · 2024-11-22
> **Response to Reviewer uXiv**
>
> We thank the reviewer for their positive feedback and for recognizing Chimera as a fundamental model architecture that demonstrates strong performance across vision, graph, and natural language processing tasks, as well as it being **a significant contribution to the ML community**. Below, we address their questions and concerns:
>
> > 1. Why are the number of layers in Chimera 2x that of ViT? Is this a fair comparison?
>
> The concept of a "layer" differs between Transformers and SSMs. In Transformers, a layer consists of an Attention block and an FFN block. In Mamba, a layer corresponds to a single MambaBlock. A Transformer block contains twice the number of parameters as a MambaBlock and **therefore the effective number of layers in both the models are the same** with two SSM layers being equivalent to one Transformer layer.
>
> For example, Pythia-2.8B [1] uses 32 Transformer layers, while Mamba-2.7B [2] employs 64 S6 layers—resulting in equivalent parameter counts and FLOPs. Hence, this is a fair comparison, as both models are controlled for computational cost and aligns with community standards for evaluating SSMs.
>
>
> > 2. Can authors include a discussion on the computational efficiency of Chimera?
>
> We refer the reviewer to sections 1A, 1B, 1C of the global response for a discussion about the computational efficiency of Chimera.
>
>
> > 3. Do larger Chimera models underperform compared to smaller models? Can the authors verify it on NLP and CV tasks.
>
> In our experience, larger Chimera models consistently outperform smaller models across CV and NLP tasks. Specifically,
>
> **a. NLP Task**
>
> On the GLUE task, Chimera (DAG) with 110M parameters achieves a GLUE score of 83.93%, outperforming the smaller 70M model with a GLUE score of 80.6%.
>
>
> | Method          | # Params | C4 Pretrain L𝑐𝑒 | C4 Pretrain Acc (%) | GLUE |
> |------------------|----------|------------------|----------------------|--------------|
> | Chimera (DAG)             | 70M      | 1.68             | 65.6                 | 80.6         |
> | Chimera (DAG)   | 110M     | 1.46             | 68.9                 | 83.93        |
>
>
> **b. CV Task**
>
> On the ImageNet-1k classification task, Chimera's 88M parameter model achieves 81.4% Top-1 accuracy and outperforming the 22M parameter model, which achieves 77.8% Top-1 accuracy.
>
>
> | Method          | # Params | Top-1 Acc (%) | Top-1 Acc EMA (%) | Top-5 Acc (%) | Top-5 Acc EMA (%) |
> |------------------|----------|---------------|-------------------|---------------|-------------------|
> | Chimera-ViT-S    | 22M      | 77.8          | 76.7              | 93.9          | 93.5              |
> | Chimera-ViT-B   | 88M      | 81.4          | 82.1              | 95.4          | 95.9              |
>
>
> > 4. Is the optimization landscape of Transformers (ViT/BERT ) better than that of SSMs (Chimera). Could the authors add a figure on training loss to compare them?
>
>
> In our experiments, we did not observe any deterioration in the optimization landscapes of Chimera.  We have attached the training loss curves and the eval metrics curves for BERT-B (110M) and Chimera (DAG, 110M) in the supplementary.
>
> ----
> ----
>
> ### References
> [1]: Albert Gu, Karan Goel, and Christopher Ré. Efficiently modeling long sequences with structured state spaces. ICLR, 2022.
>
> [2]: Stella Biderman et al. Pythia: A Suite for Analyzing Large Language Models Across Training and Scaling

---

> > ### Comment · Reviewer_uXiv · 2024-11-22
> >
> > I appreciate the authors' efforts in their rebuttal and they have addressed most of my questions. I have two follow-up questions:
> >
> > 1. I understand that Mamba models typically have twice the depth of Transformer models in order to maintain the same number of parameters. However, Mamba-series papers often receive criticism for this practice. I believe it is necessary to conduct an experiment that ensures the same depth. The authors could increase the width to maintain the same number of parameters.
> >
> > 2. While having a theoretical value of computational complexity is beneficial, an empirical comparison with Transformer models is also necessary and would strengthen the comparison.

---

> > > ### Comment · Reviewer_uXiv · 2024-11-26
> > >
> > > I would appreciate it if the authors could respond before the deadline.

---

> > > > ### Author Response · Authors · 2024-11-26
> > > >
> > > > We apologize to the reviewer for the delay; we are waiting on GPU resources to proceed with pre-training the requested models. We are actively working on addressing your comments and we sincerely appreciate your patience.

---

> > > > > ### Author Response · Authors · 2024-12-03
> > > > >
> > > > > We thank the reviewer for their patience and we kindly refer them to the global "Additional Experiments" response.

---

### Official Review · Reviewer_nw18 · 2024-11-04

**Soundness:** 4
**Presentation:** 3
**Contribution:** 3
**Rating:** 6
**Confidence:** 3

**Summary:**

This paper introduces an SSM framework that can be adapted to several data topology. Moreover, they propose an efficient way of processing the topology when the structure is in the form of a DAG, together with a technique to decompose common modalities such as images and text into DAG components. The method is tested on 3 different modalities including image, text and graph benchmarks.

**Strengths:**

- Incorporating the data topology in the SSM architecture via the resolvent of the adjacency matrix is a useful tool for extending SSM models into graph community. While most of the existing approaches are flattening the graphs and rely on the expressivity of positional encodings to directly apply the SSM on a sequence, CHIMERA are efficiently incorporate the graph structure via adjacency matrix masking.
- The fact that, when the structure is constrained to be a line graph,  the model correspond to the existing sequence-based SSM,  guarantees the good performance beyond graph domain.
- The decomposition of existing structure into DAG components improves the efficiency, an important advantage for SSM models over existing architectures.
- The experimental results show competitive performance with reduced complexity.

**Weaknesses:**

- The paper claims to propose a method that is able to deal with several types of data. However, the core contribution is integrating a graph structure into the SSM, leveraging the fact that sequences  can be represented as linear graphs, while images can be represented as grid-like graphs. Based on this well-known fact, any graph architecture can be (over)stated as being apliable across modalities.
- The operator proposed in Equation(12) is wildly used in spectral-based graph neural networks. Rediscovering it via an SSM lens and using specific techniques to compute it efficiently is an important contribution. However, discussing its relationship with attention-based GNNs (like GAT and Graph Transformers) or efficient GNN variants would provide a more comprehensive overview of the field, extending the context beyond SSMs
- A key advantage of employing SSMs consist in reducing complexity and creating more efficient computation.  While the paper does a good job on motivating linear complexity (as opposed to the quadratic one used by models such as BERT), it fall short on evaluating the efficiency experimentaly. Reporting the empirical time complexity would help better emphasize the advantages of using CHIMERA compared to existing models.

**Questions:**

Please see the sections above.

---

> ### Author Response · Authors · 2024-11-22
> **Response to Reviewer nw18**
>
> We thank the reviewer for their positive review and for their appreciation of Chimera’s ability to efficiently incorporate graph structure without relying on positional encodings, as well as its strong performance beyond graphs. We address their concerns below:
>
> > 1. Can any graph architecture can be (over)stated as being applicable across modalities?
>
> We acknowledge the reviewer's perspective, but we would like to clarify why the claim of applicability across domains is warranted for Chimera and not so much for other graph methods. Chimera uniquely demonstrates state-of-the-art performance across language, image, and graph tasks using the same unified architecture. This is unlike methods like GCN and GAT, which, despite their similarity to CNNs, often fail to surpass CNN-based baselines on image tasks, since they suffer from oversmoothing as the number of layers increase [1, 2]. This consistency in strong performance without requiring domain-specific modifications suggests that Chimera is a step in the right direction in effectively capturing the inductive bias of topology.
>
> > 2. Can the authors compare Chimera against attention-based GNNs (like GAT and Graph Transformers) or efficient GNN variants?
>
> *a. Comparing Chimera and Attention-based GNNs*
>
> Among attention-based methods, GAT [2] employs attention locally on neighboring nodes, making it less effective at capturing long-range dependencies. Graph Transformers [3] address this by using topology-agnostic global attention but heavily rely on positional encodings to model graph topology. GPS-Transformer [4], which is a more recent iteration, combines local message-passing with global attention but still depends on positional encodings to capture graph structure.
>
> In contrast, Chimera adopts a fundamentally different approach: It generalizes Mamba with its selection mechanism for long-range dependencies into directly incorporating the underlying topology via the resolvent operator. This approach not only eliminates the need for position embeddings but also achieves state-of-the-art performance across diverse domains, unlike graph-specific methods.
>
> *b. Comparing Chimera and Efficient GNNs*
>
> Efficient GNN variants, such as Linear GCN [1], aggregate features across $k$-hop neighborhoods by collapsing $k$ layers into a single operation, $H = \hat{A}^k X$, where $\hat{A}$ is the normalized adjacency matrix. However, Chimera differs in key aspects:
>
> 1. It aggregates features per layer, combining global context at each step rather than merely expanding the receptive field as in Linear GCN.
> 2. Unlike Linear GCN's unweighted adjacency matrix, Chimera parameterizes $\hat{A}$ as a generalization of Mamba-2 with its selection mechanism for modeling long-range dependencies.
> 3. Chimera models its outputs as $(L \odot CB^T)V$ (Sec 3.3), integrating graph topology within a Transformer like key-query-value framework which is widely recognized for strong performance across domains.
>
> These differences allows Chimera to achieve state-of-the-art performance across a wide array of modalities whereas  Linear GCN is primarily used for graph data.
>
> We thank the reviewer for highlighting this and will include these comparisons in the next version of our paper.
>
> > 3. Can the authors include a discussion on the computational complexity of the method?
>
>  We refer the reviewer to Sections 1A,B,C of the global response where we have included a discussion on the computational cost of Chimera.
>
> ----
> ----
>
> ### References
> [1] Rusch, T. Konstantin, Michael M. Bronstein, and Siddhartha Mishra. "A survey on oversmoothing in graph neural networks." arXiv preprint arXiv:2303.10993 (2023).
>
> [2] Zhao, Lingxiao, and Leman Akoglu. "Pairnorm: Tackling oversmoothing in gnns." arXiv preprint arXiv:1909.12223 (2019).
>
> [3] Cai, Chen, and Yusu Wang. "A note on over-smoothing for graph neural networks." arXiv preprint arXiv:2006.13318 (2020).
>
> [4]: Wu, F., Souza, A., Zhang, T., Fifty, C., Yu, T., & Weinberger, K. (2019, May). Simplifying graph convolutional networks. In International conference on machine learning (pp. 6861-6871). PMLR.
>
> [5]: Petar Veličković, Guillem Cucurull, Arantxa Casanova, Adriana Romero, Pietro Liò, Yoshua Bengio. Graph Attention Networks.
>
> [6]: Seongjun Yun, Minbyul Jeong, Raehyun Kim, Jaewoo Kang, Hyunwoo J. Kim. Graph Transformer Networks.
>
> [7]: Ladislav Rampášek, Mikhail Galkin, Vijay Prakash Dwivedi, Anh Tuan Luu, Guy Wolf, Dominique Beaini. Recipe for a General, Powerful, Scalable Graph Transformer.

---

> > ### Comment · Reviewer_nw18 · 2024-11-27
> >
> > I appreciate the authors' effort in addressing my review.
> >
> > Overall I am happy with the comparison against efficient/attention-based GNN methods and the clarification regarding the applicability of the model across modalities.
> >
> > Regarding the efficiency, I still have some concern regarding the empirical complexity of the model. I understand the difficulties in developing a dedicated kernel which is not the focus of the current work. However, the efficiency aspect is mentioned several times during the manuscript which can create confusions. I believe these limitations and the need for a faster implementation needs to be clearly stated in the paper.
> >
> > This is the reason why I still recommend acceptance, but I will maintain my score.

---

### Official Review · Reviewer_yGvr · 2024-11-04

**Soundness:** 3
**Presentation:** 3
**Contribution:** 3
**Rating:** 6
**Confidence:** 4

**Summary:**

The paper proposes Chimera, a framework that generalizes state space models (SSMs) to handle arbitrary data topologies by reformulating them through resolvent operations on adjacency matrices. The paper introduces a principled mathematical approach to incorporating topology into SSMs, achieving competitive results across various tasks. However, the method faces practical limitations for general topologies where DAG assumption does not hold and approximations are required. In addition, the discussion is lacking if the method can also generalize to more structured topologies such as Lie groups.

**Strengths:**

1. The paper potentially expands the domain of application of recent state-space models (with SMA form) beyond plain sequences.
2. Theorem 4.5 and the subsequent sections on optimized forward and backward pass implementation make the proposed method the most efficient for DAG-topologies.
3. The normalization scheme is well-motivated and theoretically grounded. I can see it is an essential ingredient for applying Chimera to the wide range of topologies.
4. Experiments demonstrate clear advantages of the proposed Chimera model in comparison with transformer, mixer, M2 and Hyena architectures.

**Weaknesses:**

1. The insight of SSMs operating on directed line topology is obvious and I suppose trivially holds for any sequence model.
2. Figure 1 does not contribute much to the narrative in the paper. I suggest the authors make a figure more specific to the method described in the paper.
3. While the proposed method claims to generalize SMA-SSMs to any topology, I am not sure it can handle topological groups such as Lie groups. The paper would benefit from additional discussion on the possible extension of SSMs to group-structured topologies given the rising interest in alternatives to self-attention for group-equivariance [1,2,3 etc].
4. While in theory, the proposed approach applies to a wide range of topologies, the practical application requires either a strong assumption like DAG or approximating L as in Eq.19 which can be costly for large graphs. It limits the practical applicability of the proposed model, especially given the results in Table 5 that demonstrate the exact version to work much better than the approximated one.
5. The paper will benefit from a more detailed discussion on computational complexity in the general case when the DAG assumption does not hold.
6.  While I understand the motivation of setting k=diam(G), the paper would benefit from the ablation study on how the performance depends on k when k < diam(G). This ablation study aims to verify if the method can be optimized for large graphs with large diam(G).

[1]. D. Liang et al. PointMamba: A Simple State Space Model for Point Cloud Analysis. NeurIPS 2024.
[2]. A. Moskalev et al. SE(3)-Hyena Operator for Scalable Equivariant Learning. GRaM@ICML 2024.
[3]. Y. Schiff et al. Caduceus: Bi-Directional Equivariant Long-Range DNA Sequence Modeling. ICML2024.

Minor:
 - line 161 footnote typos
 - Notation clash in Proposition 4.1: consider changing A^T to A^{|V|}, otherwise, A^T reads as transposed.

**Questions:**

Questions:
 - Is it possible to extend the proposed method to incorporate edge features when provided? I realize that the proposed method is a first step towards topology-aware SSMs, a brief discussion on the possibility of incorporating edge features will be useful.
 - Why are the results in Tables 1,2,3 reported only over one data split?

---

> ### Author Response · Authors · 2024-11-22
> **Response to Reviewer yGvr**
>
> We thank the reviewer for their positive feedback and for recognizing Chimera as a principled method with well-motivated, theoretically grounded method normalization as well as its competitive results across various tasks. We now address their concerns:
>
> > 1. The insight of SSMs operating on directed line topology is obvious and holds trivially.
>
>
> We agree that the idea of SSMs operating on line graph topologies due to their recurrence is indeed intuitive—this was the starting point for our work as well. However, Chimera is **the first work to formalize this connection mathematically**, which is a crucial step that enables the generalization of SSMs beyond sequences to general graph topologies across domains.
>
> > 2. Could the figure 1 be improved to reflect the method?
>
> We appreciate the reviewer's suggestion and we will provide a clearer illustration of our approach in the next revision.
>
>
> > 3. Can Chimera handle Lie groups?
>
> Chimera currently focuses on modeling data topologies in the graph sense and does not address equivariance or invariance to specific group operations.
>
> We believe the confusion arises from the term "topology," which is overloaded in the literature. In this work, we use topology in the graphical sense, treating nodes as abstract objects connected by edges that represent neighborhood relationships. This is in line with past works such as GraphGPS+Transformer [1], GINE [2], and GatedGCN [3]. We chose the term "topology" to emphasize that Chimera is not limited to traditional graph data but is designed for any modality with an underlying notion of neighborhoodness, such as images, language, and audio.
>
> We believe the reviewer may be referring to a different line of work where topology is understood in the algebraic sense, as in LieTransformer [5] and SE(3)-Hyena [6] that focuses on equivariance or invariance to group operations,. Nevertheless, we note that since Chimera operates on the underlying graph topology, it is inherently invariant to transformations that preserve the graph structure, such as permutations of nodes.
>
> We appreciate the reviewer’s feedback and welcome their suggestions on how to make this distinction clearer. While we currently rely on context to differentiate these meanings, we are happy to revise the manuscript for greater clarity.
>
>
> > 4a. For general graphs is the SOTA achieved via the exact cubic method. And the exact version to work much better than the approximated one which limits practical utility.
> >
> > 4b. Can the authors include a discussion on the computational complexity of the method?
>
> a. The reviewer is correct that the earlier state-of-the-art results for general graphs were based on the exact cubic method. However, we note that we have since improved the approximation method, which now surpasses all baselines and achieves performance on-par with the cubic method on Peptides-Func and Peptides-Struct, effectively mitigating the cubic complexity of the exact method. We refer the reviewer to Section 2 of the global response for details.
>
> b. We have included a discussion on the computational cost of Chimera in Sections 1A,B,C of the global response.
>
> > 5. What happens when k < dia(G) in the approximation method?
>
> We thank the reviewer for suggesting this ablation and we have now included it in our response below. Our results show that the model is robust to using smaller values of k. Additionally, we note that the computational cost of Chimera has a weak dependence on k, **scaling only logarithmically with the graph diameter**.
>
>
> **Setting:** We evaluated the effect of using smaller values of approximation degree k on the Peptides-Struct dataset, with k = Dia(G)/factor, and report MAE over one seed due to time constraints:
>
> | Factor | MAE (↓)   |
> |------------------------------|-----------|
> | 1                   | 0.24208   |
> | 2                  | 0.24409   |
> | 3                  | 0.24419   |
> | 4                | 0.24408   |
>
> We observe a small drop in performance as factor increases from 1 to 2, but overall the results are robust to smaller values of k. We believe that this is an important ablation and we will definitely include it in the next revision of our manuscript.
>
> > 6. Is it possible to extend the proposed method to incorporate edge features when provided?
>
> We apologize for not including this detail in the current manuscript, and we would like to clarify that our reported SOTA results already incorporate edge features using the following approach. Specifically, in equation 11 in the paper, we add edge features as an additional feature to the calculation of $A_{ij}$ where,
>
> $A_{i, j} = \frac{\exp(-\Delta_i) + \exp(-\Delta_j) + \exp(-\Delta_e)}{3}$
>
> where $\Delta_e = \text{Softplus}(f_e(E))$ and $E$ are the edge features, and $f_e$ is a projection. We will update the manuscript with this detail in the next revision.

---

> > ### Author Response · Authors · 2024-11-22
> > **Response to Reviewer yGvr, Part 2**
> >
> > > 7. Why are the results in Tables 1,2,3 reported only over one data split?
> >
> >
> > Tables 1, 2, and 3 report results on large-scale language and vision datasets, where **it is a community standard to only present validation scores** from a single training run. This due to the low variance of large models (e.g., 110M for language, 88M for vision) on the validation.
> >
> > In contrast, for graph datasets like LRGB, the community standard is to report mean and variance on the test split across multiple runs as the models are smaller models (~500k parameters) and are more sensitive to random seeds.
> >
> > > 8. Typos and notation clash in Proposition 4.1
> >
> > We will surely fix the footnote typos and address the notation clash in Proposition 4.1 by changing $A^T$ to $A^{|V|}$ in our next revision.
> >
> >
> > ----
> > ----
> >
> > ### References
> >
> > [1]: Ladislav Rampášek, Michael Galkin, Vijay Prakash Dwivedi, Anh Tuan Luu, Guy Wolf, and Dominique
> > Beaini. Recipe for a general, powerful, scalable graph transformer.
> >
> > [2]: Weihua Hu, Bowen Liu, Joseph Gomes, Marinka Zitnik, Percy Liang, Vijay Pande, and Jure Leskovec.
> > Strategies for pre-training graph neural networks.
> >
> > [3]: Xavier Bresson, Thomas Laurent. Residual Gated Graph ConvNets.
> >
> > [4]: Vijay Prakash Dwivedi, Ladislav Rampášek, Mikhail Galkin, Ali Parviz, Guy Wolf, Anh Tuan Luu, Dominique Beaini. Long Range Graph Benchmark.
> >
> > [5]: Michael Hutchinson, Charline Le Lan, Sheheryar Zaidi, Emilien Dupont, Yee Whye Teh, Hyunjik Kim. LieTransformer: Equivariant self-attention for Lie Groups.
> >
> > [6]: Artem Moskalev, Mangal Prakash, Rui Liao, Tommaso Mansi. SE(3)-Hyena Operator for Scalable Equivariant Learning.
> >
> > [7]: R. Ramakrishnan, P. O. Dral, M. Rupp, O. A. von Lilienfeld, Quantum chemistry structures and properties of 134 kilo molecules, Scientific Data 1, 140022, 2014.

---

> > > ### Comment · Reviewer_yGvr · 2024-11-28
> > > **Response to authors**
> > >
> > > I appreciate the authors' effort to address the weaknesses. The authors addressed most of my concerns.
> > >
> > > However, I agree with the rest of the reviewers that empirical runtime (and preferably peak GPU memory consumption) comparison against other methods is necessary. Ideally, authors could provide empirical runtime and memory consumption (with respect to the increasing number of nodes or radius graph) of their method on the task that requires approximating the resolvent, and compare it against runtime/memory of other baselines (e.g. transformer-based).
> > >
> > > I think this experiment can significantly support the efficiency claim of the paper, which is one of the cornerstones of the work. I trust the authors will address this, and I will keep the current score, which is already positive, for now. Based on the presence (or absence) of this experiment, and discussion with other reviewers, I will consider raising or lowering the score.

---

> > > > ### Author Response · Authors · 2024-12-03
> > > >
> > > > We thank the reviewer for their patience and we kindly refer them to the global "Additional Experiments" response.

---

> > > > > ### Comment · Reviewer_yGvr · 2024-12-03
> > > > >
> > > > > I appreciate authors provided a runtime comparison of their method. I think the presented runtime analysis is limited since the authors do not provide runtime with respect to the increasing number of nodes or radius graph, which makes it hard to understand the overall scaling (in terms of compute) behavior of the model. Nevertheless, I am convinced that even the presented preliminary analysis already makes the paper stronger. I will maintain my score, which is already a positive assessment.

---

### Official Review · Reviewer_eUvy · 2024-11-06

**Soundness:** 2
**Presentation:** 3
**Contribution:** 3
**Rating:** 5
**Confidence:** 4

**Summary:**

This paper studies state-space models for data beyond sequences. Starting from an observation that connects the mask matrix in Structured Masked Attention and the resolvent of the adjacency matrix, this paper proposes to utilize this observation to inject the topological structure of data into SSMs such that the SSMs can deal with data structures beyond sequences, e.g., images and graphs. This is done by parametrizing the "adjacency matrix" accordingly for each data structure. Multiple tricks are also proposed to improve the accuracy and efficiency of computing the resolvent, including normalization for better numerical conditions and approximation for faster speed. Experiments have demonstrated its great performance.

**Strengths:**

1. This paper is well-written and easy to follow.
2. The observation is interesting and makes sense.
3. The implementation (i.e., those tricks for improving Chimera) is straightforward.
4. The empirical results look good.

**Weaknesses:**

1. I know many experiments have been done since this paper claims contributions over multiple data structures, but I do want to see if Chimera can outperform those SSMs proposed in their respective domains, e.g., mamba2 for NLP tasks, vision mamba for CV tasks, etc.
2. The efficiency analysis is not enough: a detailed speed benchmark and description of the way of computing resolvent in practice are needed.
    - Proposition 4.4 involves the use of Gaussian elimination,  whose efficiency may highly depend on practical cases. Could the author clarify what implementation is used in practice to achieve guaranteed linear complexity?
    - If I'm not mistaken, when dealing with graph-structured data, the claimed SOTA performance is achieved by computing the resolvent exactly, right? Does this take cubic complexity? And when using the approximation trick for better efficiency, actually the performance is largely decayed, which is no better than the previous graph mamba works in most cases.
    - BTW, I'm curious about how things are implemented exactly. I was wondering if the authors plan to release the code.
3. How can the model deal with edge features for graph-structured data?
4. It seems to me this work is similar in spirit to another recent graph SSM work [1], although starting from a different story. For example, 1) their LapPE and the resolvent in Chimera for global dependency; 2) their computation can be written similarly as Eq. (5) in Chimera to achieve linear complexity while capturing global dependency; 3) their local + global module v.s. the global module and the local convolution in Chimera. Their work is so recent, so I believe the comparison is not required. But it would be great if the authors could provide a discussion of this work and Chimera.

My major concern is about the practical efficiency, which I think is not presented comprehensively yet in the current manuscript. I believe at least a speed benchmark is required. Overall, I like this paper and I'm more than willing to increase the score if this concern can be resolved.

[1] Huang, Yinan, Siqi Miao, and Pan Li. "What Can We Learn from State Space Models for Machine Learning on Graphs?." arXiv preprint arXiv:2406.05815 (2024).

**Questions:**

See above.

---

> ### Author Response · Authors · 2024-11-22
> **Response to Reviewer eUvy**
>
> We are glad that the reviewer found Chimera interesting and they recognized its strong empirical performance. We now address the concerns raised by the reviewer.
>
>
> > 1. Can Chimera outperform Mamba2 on NLP tasks and Vision Mamba on CV tasks.
>
> We note that in the causal language modeling setting with a directed line graph topology, **Chimera reduces to Mamba2** (Prop 3.3), and it therefore has an equivalent performance to Mamba2 on causal NLP tasks. Furthermore, on bidirectional NLP tasks, we show that Chimera outperforms attention-based BERT baseline by a GLUE score of 0.73 (Table 1)
>
> We cannot directly compare the results between Chimera and Vision Mamba because Chimera uses the standard training recipe used in ViT [5], Monarch Mixer [4] whereas Vision Mamba uses the DeiT [3] training recipe, introduced for distillation-based models. However, we note that the underlying module of Vision Mamba corresponds to the Fwd&Rev setting in our Table 3 ablation, where **Chimera's 2D-DAG structure demonstrates superior performance**.
>
> >  2a. How is the DAG Decomposition method implemented in practice?
> >
> >  2b. For general graphs is the SOTA achieved via the exact cubic method? The approx method has decayed performance.
> >
> >  2c. How are other variants namely the exact method, the approx method and the DAG method implemented?
> >
> >  2d. Would the authors provide code?
> >
> >  2e. Concerns about the practical efficiency.
>
>
> We kindly refer the reviewer to our global response for a detailed treatment of the computational aspects of Chimera. For convenience we summarized some of our responses here:
>
> a) The DAG Decomposition method is implemented as follows:
> 1. For line graph topology, we use Mamba2 kernels with a linear time complexity to implement the DAG decomposition method.
> 2. For the general DAG method, we use the fast squaring trick, which, as the reviewer correctly noted, is more hardware-friendly than Gaussian elimination. While our method has linear theoretical complexity, in practice, the fast squaring trick is more efficient for the current datasets. However, for sufficiently large graphs, Gaussian elimination might be faster.
>
> b) The reviewer is correct that the earlier state-of-the-art results for general graphs were based on the exact cubic method. However, we note that we have since improved the approximation method, which now achieves performance on-par with the cubic method on Peptides-Func and Peptides-Struct, effectively mitigating the cubic complexity of the exact method. Please refer to Section 2 in the global response for details.
>
> c) The implementation details for the exact, approximate, and DAG methods are provided in Section 1A of the global response.
>
> d) We will release the code in the next revision of our paper to ensure transparency and reproducibility.
>
> e) We address practical efficiency in Sections 1B and 1C of the global response
>
> > 3. How can the model deal with edge features for graph-structured data?
>
> We apologize for not including this detail in the current manuscript, and we would like to clarify that our reported SOTA results already incorporate edge features using the following approach. Specifically, in equation 11 in the paper, we add edge features as an additional feature to the calculation of $A_{ij}$ where,
>
> $A_{i, j} = \frac{\exp(-\Delta_i) + \exp(-\Delta_j) + \exp(-\Delta_e)}{3}$
>
> where $\Delta_e = \text{Softplus}(f_e(E))$ and $E$ are the edge features, and $f_e$ is a projection. We will update the manuscript with this detail in the next revision.

---

> > ### Author Response · Authors · 2024-11-22
> > **Response to Reviewer eUvy, Part 2**
> >
> > > 4. Could the authors compare against the very recent work Graph State Space Convolution (GSSC) which is in a similar in spirit?
> >
> > We thank the reviewer for highlighting the recent work, Graph State Space Convolution (GSSC) [1], which is indeed aligned in spirit with Chimera. While both approaches share similarities, there are a few key differences that we now outline:
> >
> > 1. **Motivation: Chimera generalizes SSMs, while GSSC builds position embeddings**:
> > As the reviewer correctly observed, Chimera and GSSC stem from different motivations, which also inform their respective parameterizations. Chimera is motivated as a unified framework to generalize SSMs like Mamba across domains with graph topologies, avoiding the need for designing domain-specific position embeddings. On the other hand, GSSC utilizes graph Laplacian eigenvectors as position embeddings specifically for graph datasets, focusing on capturing the structure of the graph.
> > 2. **Methodology: On a line graph, Chimera is Mamba, GSSC is Linear Attention [2]**:
> > These differing motivations result in distinct parameterizations and models. Chimera parameterizes the adjacency matrix \(A\) to make it data-dependent, whereas GSSC relies on the unweighted adjacency matrix along with node features. This leads to fundamentally different methods. For example, in the case of line graph topologies, Chimera reduces to Mamba, while GSSC reduces to linear attention [2] (Eq 3,6). It is well established that Mamba outperforms linear attention on large-scale datasets like language and images, enabling Chimera to achieve state-of-the-art results across these modalities, unlike GSSC.
> > 3. **Cost: Chimera and GSSC share the same computational complexity**:
> > Both approaches employ strategies to mitigate cubic complexity. GSSC achieves this by truncating eigenvalue decomposition to the top \(d\) eigenvalues, while Chimera achieves this by powering up the series approximation to an approximate value. However, in both cases, fully computing their respective operations would lead to cubic complexity in the number of nodes.
> >
> > We once again thank the reviewer for bringing this recent work to our attention and will include a detailed comparison in next revision of our paper.
> >
> > ----
> > ----
> >
> > ### References
> >
> > [1]: Huang, Yinan, Siqi Miao, and Pan Li. "What Can We Learn from State Space Models for Machine Learning on Graphs?." arXiv preprint arXiv:2406.05815 (2024).
> >
> > [2]: Angelos Katharopoulos, Apoorv Vyas, Nikolaos Pappas, François Fleuret. Transformers are RNNs: Fast Autoregressive Transformers with Linear Attention
> >
> > [3]: Hugo Touvron, Matthieu Cord, Matthijs Douze, Francisco Massa, Alexandre Sablayrolles, Hervé Jégou. Training data-efficient image transformers & distillation through attention
> >
> > [4]: Daniel Y. Fu, Simran Arora, Jessica Grogan, Isys Johnson, Sabri Eyuboglu, Armin W. Thomas, Benjamin Spector, Michael Poli, Atri Rudra, Christopher Ré. Monarch Mixer: A Simple Sub-Quadratic GEMM-Based Architecture
> >
> > [5]: Alexey Dosovitskiy, Lucas Beyer, Alexander Kolesnikov, Dirk Weissenborn, Xiaohua Zhai, Thomas Unterthiner, Mostafa Dehghani, Matthias Minderer, Georg Heigold, Sylvain Gelly, Jakob Uszkoreit, Neil Houlsby. An Image is Worth 16x16 Words: Transformers for Image Recognition at Scale

---

> ### Comment · Reviewer_eUvy · 2024-11-22
>
> Thanks for the response.
>
> I believe a detailed benchmark on the inference speed of Chimera compared with other efficient baselines is required for publication, e.g., Seq length v.s. Time, like the Fig. 10 in Mamba2's paper. Otherwise, you can hardly claim this is still an efficient model given that Chimera involves many computations that do not occur in the original Mamba or efficient transformer architectures.
>
> In addition, correct me if I'm wrong, given your response, I'm concerned/confused that if you cannot implement this method with linear complexity in practice, how could you really claim efficiency in this manuscript? And if this model is not efficient practically, then it essentially has lost one of the core benefits of SSMs or Mamba, that is, efficiency? The paper of Mamba also includes some tricky computation, and one reason they can be accepted (eventually) is that they also propose a way to implement that tricky computation with linear complexity. And even they have included such implementation, they were rejected at ICLR. Therefore, it seems to me it should also be required for this manuscript to provide an efficient implementation of Chimera for acceptance. Otherwise, it can be hard to vote for acceptance of this manuscript, because it sounds a bit weird if one paper claims to extend SSMs/Mamba but essentially cannot implement their methods with linear complexity like SSMs/Mamba and have claimed efficiency in their manuscript. And if there is no computational gain in Chimera, shouldn't one expect it to be compared with more advanced transformer models, like those LLMs?
>
> Please do correct me if I misunderstand anything. If I'm not mistaken and the practical efficiency concern cannot be resolved, I may have to lower my rating.

---

> > ### Author Response · Authors · 2024-11-25
> > **Response to Reviewer eUvy**
> >
> > We deeply appreciate the reviewer's continued engagement with our work. As we understand it, **the reviewer’s main concern is that Chimera must have a practically efficient implementation at the time of submission to be scientifically valuable**. We would like to address this perspective from a broader point of view before responding to specific concerns in detail.
> >
> > We completely understand the reviewer’s viewpoint and agree that highly impactful methods, such as the Mamba series, generally combine methodological contributions with optimized implementations. Similarly, we acknowledge that Chimera would benefit from an optimized implementation, which could further enhance its practicality.
> >
> > However, we believe that methods can hold significant scientific value by introducing **new design principles and advancing core modeling understanding**, even in the absence of immediate optimized implementation. Many foundational approaches in machine learning have followed a similar trajectory, where impactful modeling innovations were introduced first, and fast implementations were developed later. For instance, the attention mechanism first demonstrated strong modeling potential [1], while years later subsequent work, such as FlashAttention [2], provided algorithmic and implementation improvements for substantially increased efficiency.
> >
> > Chimera is also designed with the co-evolution of modeling principles and efficient algorithms in mind. While the focus of this work is on introducing **methodological innovations towards a unified model capable of handling domains with arbitrary graph topologies**, it is also designed so that there exist avenues to much more optimized implementations, some of which are outlined below. While we fully agree that Chimera would benefit from faster implementations, we felt it is beyond the scope of the current submission; we note that Chimera is a strong generalization of Mamba-2, which itself required deep systems expertise and over 6400 lines of Triton code in its optimized implementation. We are optimistic and excited about future work that continue to improve the implementations of our ideas.
> >
> >
> > We now expand on these arguments below, and ultimately request the reviewer to consider our work with the framing: **“Would publishing this method benefit the scientific community?”**
> >
> > ----
> > ----
> >
> > ### 1. Chimera's opens up new directions for domain-agnostic learning
> >
> > Chimera's **methodology provides valuable scientific insights** beyond efficient implementation, which we believe will greatly benefit the community:
> >
> > *A. Chimera proves that exists a single unified model that works well across domains.*
> >
> > Instead of relying on domain-specific models like Mamba-2 or BERT for language, ConvNeXT or ViT for vision, and GPS or GraphMamba for graphs, Chimera is, to the best of our knowledge, the only model that works across domains out-of-the-box.
> >
> > *B. Chimera proves that topology is a strong inductive bias across domains.*
> >
> > Chimera shows that topology serves as a powerful and universal inductive bias, and that its methodology leverages this bias effectively. This mitigates the reliance on domain-specific methods and modifications that are typically used in the community.
> >
> > ---
> > ---
> >
> >
> > ### 2.  Chimera solves a harder problem; problem dependent complexity is expected.
> >
> > > "because it sounds a bit weird if one paper claims to extend SSMs/Mamba but essentially cannot implement their methods with linear complexity like SSMs/Mamba"
> >
> > **Graph Structure-Compute Tradeoff**
> >
> > *"Harder Problem → Reduced Structure → More Compute"*
> >
> > Since we are generalizing SSMs/Mamba to a broader class of problems, it should be **expected** that Chimera be more expensive than the Mamba-2. On one end, Chimera on directed line graph (Mamba-2) models minimal node interactions and incurs a linear complexity. On the other end, Chimera on general graphs models much richer node interactions and naturally incurs a larger cost.
> >
> > Consequently, in this work we also focus on balancing expressivity with efficiency.
> > - First, we study DAGs, which lie at the intersection of linear complexity while modeling more expressivity and hence better performance than line graphs (Table 3).
> > - Second, for general graphs, we show that while the exact method has cubic complexity, the approximation method works without significant performance loss with quadratic complexity. This makes Chimera practical for modeling complex tasks like protein folding.

---

> > > ### Author Response · Authors · 2024-11-25
> > > **Response to Reviewer eUvy, Part 2**
> > >
> > > ### 3. Chimera is linear on DAGs.
> > >
> > > > "I'm concerned/confused that if you cannot implement this method with linear complexity in practice, how could you really claim efficiency in this manuscript?
> > >
> > > We would like to clarify that Section 4.1.1 explicitly presents both the algorithmic complexity (Lines 326–338), showing that Chimera can be implemented similarly to a RNN or LSTM in linear time, and the current naive vectorized implementation for training (Lines 339–362), which we use as a proof-of-concept to validate Chimera’s methodology.
> > >
> > >
> > > **Potential pathways for future optimizations**
> > >
> > > First, we reiterate that Chimera is a linear time algorithm as it can be computed by traversing the DAG, just like how an RNN or LSTM "traverses" a line graph sequentially. While this traversal is theoretically linear, it is less efficient than our vectorized implementation on GPUs. However, we note that this is not intrinsic to our method: *all RNN methods are slow unless equipped with specialized CUDA kernels*. For example, widely-used frameworks such as PyTorch and TensorFlow include dedicated fused kernels for LSTMs ([cuDNN](https://developer.nvidia.com/cudnn)) to achieve efficient performance. Similarly, designing a custom CUDA kernel for Chimera would improve runtime efficiency, and would make it truly linear time on GPUs.
> > >
> > > Second, there are additional approaches for optimizing the implementation of our method. We note that inverting $(\mathbf{I} - \mathbf{A})$ for a DAG is equivalent to inverting a lower triangular matrix, a problem well-supported by CUDA libraries like cuSPARSE. Since, using these libraries requires advanced CUDA programming, we used the simpler fast-inverse approach to validate Chimera’s scientific premises. But we believe for practitioners with CUDA expertise, leveraging these libraries is a starting path for future work.
> > >
> > > Furthermore, for the special case of grid graphs, we believe that building a dedicated kernel is a promising direction, leveraging the problem's inherent structured nature. While it is currently beyond the scope of this work and likely more involved than Mamba-2’s implementation, we view it as an exciting avenue for future research. Our results already demonstrate that preserving 2D structure significantly improves performance, marking a clear modeling advancement over 1D scan methods like Bidi-Scan [3], Cascade-Scan [3], and Windowed Scans [4], which instead simply flatten the image.
> > >
> > > ---
> > > ---
> > >
> > > ### **4. Chimera vs. Advanced Models**
> > > >"...Shouldn't one expect it to be compared with more advanced transformer models, like those LLMs?"
> > >
> > >
> > > We note that comparisons against advanced LLMs, **have already been addressed in Mamba-2 [1, Table 1]**. As proved in Proposition 3.3, Chimera acting on a directed line graph, **is Mamba-2**, and therefore, leverages the same algorithm, achieves the same empirical performance and compute efficiency.
> > >
> > > For other tasks, we have explicitly compared Chimera against state-of-the-art baselines:
> > >
> > > - **Bidirectional Language Modeling**: Chimera is compared to BERT, Monarch Mixer, FNet, where it achieves a +0.7 on GLUE over BERT (Table 1).
> > > - **Images**: Chimera is compared against ViT-B, Hyena-ViT-B, S4-ViT-B, and outperforms ViT by 2.6% on ImageNet-1k classification accuracy (Table 2).
> > > - **Graphs**: Chimera surpasses models like GCN, Gated-GCN, GINE, GPS-Transformer, GraphMamba (Table 4).
> > >
> > >
> > > ---
> > > ---
> > >
> > > ### References
> > >
> > > [1]: 2017. Ashish Vaswani, Noam Shazeer, Niki Parmar, Jakob Uszkoreit, Llion Jones, Aidan N. Gomez, Lukasz Kaiser, Illia Polosukhin. Attention Is All You Need
> > >
> > > [2]: 2022. Tri Dao, Daniel Y. Fu, Stefano Ermon, Atri Rudra, Christopher Ré. FlashAttention: Fast and Memory-Efficient Exact Attention with IO-Awareness
> > >
> > > [3]: 2024. Yue Liu, Yunjie Tian, Yuzhong Zhao, Hongtian Yu, Lingxi Xie, Yaowei Wang, Qixiang Ye, Yunfan Liu. VMamba: Visual State Space Model
> > >
> > > [4]: 2024. Tao Huang, Xiaohuan Pei, Shan You, Fei Wang, Chen Qian, Chang Xu. LocalMamba: Visual State Space Model with Windowed Selective Scan

---

> ### Comment · Reviewer_eUvy · 2024-11-26
>
> - First of all, I believe efficiency is important, especially for SSM-related methods as it is the main motivation of these methods. Being scientifically valuable does not mean it is ready for publication. And it is because of its scientific value that I'm still rating it 5. If we do not talk about efficiency at all, the key contribution of this work is mostly an observation of the topology SSMs working on, which is an interesting but actually somewhat intuitive observation.
>
> - Second, one reason that I want to lower the score is that I believe the current manuscript is in some sense overclaiming, which needs to be revised accordingly. As I said, I find it "sounds a bit weird if one paper claims to extend SSMs/Mamba but essentially cannot implement their methods with linear complexity like SSMs/Mamba and **have claimed efficiency in their manuscript**"
>     - For example, in the abstract, the authors claim "Furthermore, being topologically aware
> enables our method to achieve a linear time complexity for sequences and images". There are similar statements across the manuscript.
>     - These statements may mislead readers (like me). When I first read these statements, I believed this method was **of course** implemented with linear complexity and should be an efficient method, and it's just they'd need a speed benchmark to improve the manuscript. However, it turns out that this method can't be implemented efficiently right now, which greatly lowers the contribution of this work.

---

> > ### Comment · Reviewer_eUvy · 2024-12-02
> >
> > When I was checking the manuscript for a fair final rating, I encountered the following important questions, which may greatly impact the soundness of this paper. I hope the authors can try to resolve these questions.
> >
> > 1. Let's first not talk about empirical efficiency but theoretical complexity.
> >     - For the claimed linear theoretical complexity for **general** DAGs from Proposition 4.4, is the proof actually correct? Shouldn't the complexity be $O(nnz(A))$ for **each column**, and thus a total theoretical complexity of $O(n \cdot nnz(A))$, which of course would require quadratic algorithms both theoretically and empirically?
> >     - For general graphs, Chimera would require cubic theoretical complexity. With approximation, the theoretical complexity can only be reduced to quadratic, right?
> >
> > 2. With the essentially quadratic theoretical complexity (correct me if I'm wrong), one of course cannot have a model that can work efficiently in practice. Then, the paper seems to overclaim the contribution greatly and has never mentioned its suffering in theoretical and empirical complexity.

---

> ### Author Response · Authors · 2024-12-03
>
> We sincerely thank the reviewer for their continued thoughtful engagement with our manuscript.
>
> We would like to affirm that the statement and the proof of Proposition 4.4 are accurate. The reviewer is right that for general sparse matrices, the complexity of Gaussian elimination typically scales as $O(n \cdot nnz(I-A)$. However, for a Directed Acyclic Graph, the associated adjacency matrix is a lower-triangular matrix, and this structural property reduces the complexity of Gaussian elimination to $O(nnz(I-A))$.
>
> To illustrate this, we run Gaussian elimination by successively choosing the pivots as (i,i) for each i in the ordered list {$0,..,|V|-1$}. Observe that because the matrix is lower-triangular, when we eliminate with respect to the pivot (i,i), the remainder of row i is composed of only zeros. Therefore, the elimination operation only needs to be performed within the corresponding column instead of the entire row. This reduces the number of required arithmetic operations to $O(nnz(A[:, i]))$, where $nnz(A[:, i])$ denotes the number of non-zero elements in the column $i$. Summing over all the columns, the total complexity becomes $O(\sum_{i}^n nnz(A[:, i])) = O(nnz(A))$. In the context of Proposition 4.4, where the adjacency matrix of the DAG has $O(|V| + |E|)$ non-zero elements, we get the final claim in our proposition that the complexity is $O(|V| + |E|)$.
> We also note that for our motivating example of Mamba, the $I-A$ matrix contains exactly $2|V|$ non-zero entries, and by our theorem has a linear time complexity.
>
> We will add these details in the original proof of Proposition 4.4 and will ensure they are clearly included in the final version of the paper. Thank you for helping us clarify this important point.

---

> ### Author Response · Authors · 2024-12-04
>
> We are grateful to the reviewer for raising their concerns about the presentation of our paper, particularly regarding the complexity of the general DAG method. We recognize that the current draft may give the impression that our method is both theoretically and *practically* linear time for DAGs. In response, we have updated the manuscript with an explicit limitations section clarifying that, while the general DAG method is theoretically linear in complexity, our current GPU implementation relies on a quadratic fast-inverse trick. We also note in the limitations that developing a dedicated linear DAG kernel remains an important direction for future work, along with other potential optimizations discussed with the reviewers. We apologize for any confusion and thank the reviewer once again for their valuable feedback.

---

### Author Response · Authors · 2024-11-22
**Global Response**

We sincerely thank all the reviewers for their valuable feedback and their recognition of Chimera as a *fundamental model architecture*, with a *well-motivated design* and *theoretical grounding*. We also appreciate their acknowledgement of Chimera's *great performance on diverse data types*.

In this global response, we highlight our key contribution is a **methodological shift in modeling graph topologies, eliminating the need for ad-hoc position embeddings**. Chimera leverages SSMs for this work primarily for their ability to capture topology rather than their linear complexity. While the current PyTorch implementation has speed limitations, we believe that building dedicated CUDA/Triton kernels offers an exciting opportunity to further enhance Chimera's scalability.

Furthermore, we address the concerns about the Approximation method with improved results on Peptides Struct and Peptides Func achieved through hyperparameter tuning. The approximation method now **matches state-of-the-art performance** of the exact method, which effectively mitigates the cubic complexity of the exact method.

----
----

## 1. On Chimera's computational cost

### (A) The implementation of different variants with their costs

We recall that

- *Exact Method (Section 3.3)*: Computes the exact resolvent via matrix inversion.
- *Approximate Method (Section 4.1.2)*: Approximates the resolvent using a power series expansion up to the graph diameter.
- *DAG Method (Section 4.1)*: Decomposes graphs into multiple DAGs.


For any graph topology with the number of nodes as V and the number of edges as E:


| Method       | Theoretical FLOPs      | Our Current Implementation          | Implementation FLOPs      | Parallel Complexity (Depth) |
|--------------|-------------------------|--------------------------|----------------------------|------------------------------------|
| Exact        | $O(V^3)$               | `torch.linalg.inv`       | $O(V^3)$                  | $O(\log(V))$                      |
| Approximate  | $O(V^2 \log(dia(G)))$  | Fast squaring trick      | $O(V^2 \log(dia(G)))$     | $O(\log(dia(G)))$                 |
| DAG          | $O(V + E)$             | Fast squaring trick      | $O(V^2 \log(dia(G)))$     | $O(\log(dia(G)))$                 |


----

### \(B\) Chimera is motivated by SSMs' inherent topology rather than linearity

Chimera identifies and leverages a fundamental property of SSMs: **their ability to directly operate on line-graph topologies without relying on position embeddings**. While SSMs are often regarded for their computational efficiency compared to Transformers, this is not the primary reason why Chimera generalizes SSMs. Instead, Chimera represents a methodological advancement over prior approaches that either "flatten" data into sequences to use models like Mamba or apply topology-agnostic models such as self-attention along with domain-specific position embeddings. By leveraging this property, Chimera's unified approach generalizes across diverse modalities like language, vision, and graphs, achieving state-of-the-art results while avoiding the need for designing task-specific embeddings, such as RoPE embeddings for language [5], absolute embeddings for vision [7], or Laplacian embeddings for graphs [8].

----

### \(C\) Like Mamba, developing a fast kernel for Chimera is an important future work.


Our primary goal in this work was to introduce a methodological shift for modeling data with an underlying graph topology, with a lesser emphasis on implementation optimizations. While Chimera’s current naive PyTorch implementation is slow, there is substantial potential to reduce wall clock time through hardware-aware kernels. Developing such kernels, however, is a non-trivial task. For instance, the first version of Mamba [1], which used a parallel scan algorithm for sequence mixing, was 15x slower than FlashAttention-2 [3, 4] (state size=256, sequence length<1000) and required over 2,400 lines of CUDA code. Mamba achieved parity with FlashAttention-2 only in its second iteration, Mamba-2 (SSD) [2], which involved over 6,400 lines of Triton code developed by the creators of FlashAttention. Given that Chimera is a more involved method compared to Mamba-2, we view the development of a fast kernel as an exciting opportunity for future works.

---

> ### Author Response · Authors · 2024-11-22
> **Global Response, Part 2**
>
> ## 2. Improved results on the Long Range Graph Benchmark (LRGB)
>
> We have further tuned our experiments on the LRGB dataset, and the approximation method now achieves state-of-the-art (SOTA) performance on Peptides-Func and Peptides-Struct, **matching the exact method within standard deviations**.
>
> The updated results are summarized in the table below:
>
> | **Method**          | **Peptides Func (AP↑)** | **Peptides Struct (MAE↓)** |
> |---------------------|--------------------------|----------------------------|
> | GCN                 | 0.6860 ± 0.0050         | **0.2460 ± 0.0007**        |
> | Gated-GCN           | 0.6765 ± 0.0047         | 0.2477 ± 0.0009            |
> | GINE                | 0.6621 ± 0.0067         | 0.2473 ± 0.0017            |
> | GraphGPS            | 0.6534 ± 0.0091         | 0.2509 ± 0.0014            |
> | Chimera             | **0.7021 ± 0.003**      | **0.2460 ± 0.0002**        |
> | Chimera Approx (Prev.) | 0.6709 ± 0.0089      | 0.2521 ± 0.0006            |
> | Chimera Approx (Upd.) | **0.6979 ± 0.0057**  | **0.2420 ± 0.0013**        |
>
> Since the Approximation method achieves results on par with the exact method, it effectively alleviates the cubic complexity of the exact method and can be used instead of calculating the exact resolvant.
>
>
> The changes in the hyperparameters for both the datasets are as follows:
> | **Hyperparameter**    | **Learning Rate** | **Optimizer** | **Dropout** | **#Layers** | **Hidden Dim.** | **#Heads** | **State Size** | **MLP Layers** | **Batch Size** | **#Epochs** | **Norm**    | **Norm Style** | **MPNN** |
> |------------------------|-------------------|---------------|-------------|-------------|-----------------|------------|----------------|----------------|----------------|-------------|-------------|----------------|---------|
> | Old Config         | 0.001            | Adam          | 0.1         | 2           | 256             | 2          | 64             | 2              | 32             | 250         | BatchNorm   | PostNorm       | GCN     |
> | New Config         | 0.001            | Adam          | 0.1         | 8           | 64              | 2          | 96             | 4              | 64             | 250         | LayerNorm   | PreNorm        | GCN     |
>
> ----
> ----
> ### References
>
> [1]: Albert Gu, Karan Goel, and Christopher Ré. Efficiently modeling long sequences with structured state spaces. ICLR, 2022.
>
> [2]: Tri Dao and Albert Gu. Transformers are SSMs: Generalized models and efficient algorithms through structured state space duality. In International Conference on Machine Learning (ICML), 2024b.
>
> [3]: Tri Dao and Albert Gu. State Space Duality (Mamba-2) Blogpost [Link](https://goombalab.github.io/blog/2024/mamba2-part1-model/#the-ssd-model)
>
> [4]: Tri Dao. FlashAttention-2: Faster Attention with Better Parallelism and Work Partitioning
>
> [5]: Jianlin Su, Yu Lu, Shengfeng Pan, Ahmed Murtadha, Bo Wen, and Yunfeng Liu. Roformer: Enhanced transformer with rotary position embedding, 2023
>
> [6]: Hugo Touvron et al. Llama: Open and efficient foundation language models, 2023.
>
> [7]: Alexey Dosovitskiy et al. An image is worth 16x16 words: Transformers for image recognition at scale
>
> [8]: Ladislav Rampášek, Michael Galkin, Vijay Prakash Dwivedi, Anh Tuan Luu, Guy Wolf, and Dominique Beaini. Recipe for a general, powerful, scalable graph transformer. Advances in Neural Information Processing Systems

---

### Author Response · Authors · 2024-12-03
**Additional Experiments**

We thank all the reviewers for their thorough engagement with our work and for their insightful suggestions to improve our manuscript. We also sincerely appreciate their patience as we worked to secure the necessary resources for the suggested pretraining experiments.

Based on our understanding, a common recommendation from the reviewers is providing a wall-clock time comparison between Chimera and Transformer-based baselines. While we again emphasize that the primary focus of our work is introducing a conceptual methodological change, we fully agree that an analysis of Chimera's runtime is valuable as it would not only support Chimera's broader adoption but also inspire future directions.

To this end, we benchmark Chimera's wall-clock time against the Transformer baseline using the same architectural configurations as in our original Peptides-Struct experiment. In this realistic setting, we observe that Chimera's wall-clock time is within 1.8x of the Transformer baseline. Furthermore, we show that the wall-clock time is robust across the number of heads and model dimension. We believe that this reasonable performance offers a foundation for researchers to explore Chimera's impact on novel domains such as protein folding, where incorporating the graph topology more rigorously is essential for strong performance.

Additionally, our benchmarking identifies key optimization areas for future work; currently our implementation is intentionally simple and it incurs an overhead of about 28% on bookkeeping operations including graph-processing tasks such as converting between sparse and dense matrix formats--an overhead that can be significantly reduced through optimized implementations. Beyond this overhead, we see significant potential for optimization through dedicated kernels targeting core operations like graph inversion. For instance, in the case of directed acyclic graphs (DAGs), we believe tailored kernel implementations can reduce the complexity of graph inversion from cubic to linear time. Additionally, fusing the various operations within the block—such as local convolutions, linear projections, skip connections, and the main Chimera processing unit—offers significant optimization potential similarly to the approach used in Mamba.

Furthermore, we conduct additional experiments to isolate the effect of depth on Chimera's performance in response to reviewer uXiv's suggestion. In this experiment, we pretrain parameter-matched models: a 6-layer Chimera, a 6-layer BERT, and a 12-layer Chimera on the bidirectional language modeling task. Our results indicate that the validation performance of the two Chimera models is nearly identical, and they both significantly outperform BERT. This demonstrates that superior performance is driven by its methodological contributions rather than the increased number of layers.

We once again thank all the reviewers for their valuable suggestions, which we believe have significantly contributed to improving our manuscript.

---------
---------

### Experiment 1: Chimera's wall-clock time analysis


**A. Inference Time Experiments**
We begin by measuring the *per-layer,* *inference wall-clock time* of the approximate variant of Chimera and GPS-Transformer on the Peptides-Struct dataset. For this experiment, we adopt the architectural configurations from our state-of-the-art results--using a model dimension of 64 and 4 attention heads--ensuring parameter parity across layers for both models. We observe that the **time-ratio**, defined as $\frac{\text{time taken by Chimera}}{\text{time taken by GPS-Transformer}}$, is $\sim 1.5\times$, which we believe is reasonable for researchers to further explore Chimera’s potential across novel domains.


| **Method**    | **Wall-Clock Time Mean (ms)** | **Wall-Clock Time Std-Dev (ms)** |
|---------------|-------------------------|------------------|
| Chimera       | 11.552        | 2.831 |
| Graph-GPS     | 7.665        |  2.371|

---

> ### Author Response · Authors · 2024-12-03
>
> **B. Time-Ratio is Robust to Architectural Variations**
>
> We further conduct a series of experiments to test how the time-ratio changes by varying different hyperparameters in our model. The hyperparameters we ablate are: model dimension (d_model), number of heads, and batch-size. Note that we define the setting with `batch size = 1` setting as evaluating the computational speed of the model during "inference mode". We find that across these hyperparameters, the time-ratio between the two models remains consistent, demonstrating the robustness of our analysis to variations in architectural design and training configurations.
>
>
> *Table 1: Ablation over Model dimension*
>
> | Model Dimension | Time-Ratio |
> |---------|---------------|
> | 32      | 1.58   |
> | 64      | 1.49     |
> | 96      | 1.45   |
> | 128     | 1.49   |
>
> *Table 2: Ablation over Number of Heads*
> | Number of Heads | Time-Ratio |
> |-----------------|---------------|
> | 1               | 1.49     |
> | 2               | 1.52   |
> | 4               | 1.51   |
> | 8               | 1.53   |
>
> *Table 3: Ablation over Batch Size*
> | Batch Size | Time-Ratio  |
> |------------|---------------|
> | 1 (inference mode)         | 1.51   |
> | 2          | 1.59   |
> | 4          | 1.57   |
> | 8          | 1.58   |
> | 16         | 1.79   |
>
> ----
>
> **C. Bottleneck analysis**
>
> As we had noted in our global response, the current implementation of Chimera has a higher wall-clock time compared to Transformer-based baselines. To better understand this disparity, we analyzed where Chimera incurs the most overhead, which offers valuable insights and avenues for its optimization:
>
>
> We remark that we kept our current implementation purposefully simple and therefore a significant portion of the runtime (approximately 28%) is consumed by bookkeeping operations like converting the graph's original sparse adjacency list representation into its dense matrix form—implemented using `torch_geometric.utils.to_dense_batch`, `torch_geometric.utils.to_dense_adj`—which we use for matrix normalization, inversion and subsequent matmuls. While these conversions simplify the implementation, they are computationally expensive: Creating a dense representation on the fly involves a lot of  memory copy operations with a substantial overhead. This overhead is exacerbated by irregular batch sizes as the resulting dense matrix must accommodate the largest graph in the batch.
>
> *Future Directions*: While this work focuses primarily on validating Chimera's methodology, through the above analysis we identify pathways to significantly improving runtime performance:
>
> 1. *Preprocessing data*: Precomputing batches with similar graph sizes along with their dense representations can reduce the overhead of on-the-fly dense-to-sparse conversions.
> 2. *Direct Sparse Matrix Operations*: Using dedicated CUDA kernels from cuSPARSE can help avoid materializing dense matrices, and instead could operate directly on sparse adjacency lists to compute $My = x$, solving for $y$ without explicitly materializing $M$.
> 3. *Dedicated kernels for DAGs*: As we noted in our global response, we believe that tailored kernels for directed acyclic graphs (DAGs) can significantly reduce Chimera's practical complexity due to the structured nature of the problem.
> 4. *Fused Operations:* Fusing operations within the Chimera block—such as local convolutions, linear projections, skip connections, and the main processing unit—can further improve execution speed, similar to the optimizations used in Mamba.
>
> We believe that these optimizations represent an exciting avenue for future work. Our work showcases Chimera’s methodological promise and this analysis provides actionable insights that can help improve its runtime.
>
> -----
>
>
> **D: Wall-clock time Reduces with Coarser Approximation**
>
>
> We study the impact of the degree of approximation on wall-clock time. Specifically, we define the log-approximation *factor* such that the largest power to which the adjacency matrix $A$ is raised is determined by $\lceil \text{diameter}(G) / 2^{\text{factor}} \rceil$. As the approximation factor increases, our results indicate the expected reduction in the runtime of the fast-inverse operation relative to the full-diameter fast-inverse approximation.
>
>
> | **Log-Approx Factor** | **Ratio wrt Diameter Approximation**        |
> |--------------------|----------------------|
> | 0                  | 1.0x     |
> | 1                  | 0.88x  |
> | 2                  | 0.76x     |
> | 3                  | 0.67x     |
>
> We note that the matrix inversion currently accounts for around 15% of the total runtime, with the majority of the computational overhead arising from graph-processing operations. Nevertheless, as our prior experiments indicate that increasing the approximation factor does not significantly degrade performance, this highlights another avenue to reduce computational cost while maintaining performance.

---

> > ### Author Response · Authors · 2024-12-03
> >
> > ## Experiment 2: Chimera outperforms BERT when layers are controlled
> >
> > In this experiment, we aim to address reviewer uXiv's thoughtful question about whether Chimera's superior performance over Transformer models stems from its use of a larger number of layers, a common practice in SSM-based architectures to control the number of parameters. In this experiment, we demonstrate that Chimera retains its state-of-the-art performance even when the number of layers is controlled, highlighting that its superior performance stems from its methodological innovations.
> >
> >
> > To isolate the effect of depth, we designed an ablation study on the bidirectional language modeling task for which we consider three models:
> >
> > 1. **Chimera-12L**: 12 layers, the baseline configuration with 70M parameters.
> > 2. **BERT-6L**: 6 layers, transformer baseline with 70M parameters.
> > 3. **Chimera-6L**: 6 layers, with the expansion factor set to 4 to maintain the same parameter count as the other models.
> >
> > Due to computational constraints, we trained these models on a reduced ablation setting with 98M steps instead of the standard 245M steps. In the table below, we report the masked accuracy and cross-entropy loss of each of the pretrained models on the evaluation set.
> >
> > | **Model**          | **Masked Accuracy (↑)** | **Cross-Entropy Loss (↓)** |
> > |---------------------|---------------------|-------------------------|
> > | Chimera-12L         | **0.6363**             | 1.8142                 |
> > | Chimera-6L          | 0.6360             | **1.8108**                 |
> > | BERT-6L             | 0.6176             | 1.9466                 |
> >
> >
> > We observe that Chimera-6L and Chimera-12L achieve nearly identical performance, and both significantly outperform BERT-6L. This highlights that Chimera's superior performance is independent of increased depth, instead arising from its fundamental methodological contributions.

---

### Meta-Review · Area_Chair_whyV · 2024-12-20

**Metareview:**

**Summary:**

This paper proposes a new state-space models (SSMs) for more general data beyond sequences, including image, text and graph data. The authors devise the new method using resolvent operations on adjacency matrices, achieving strong performance with theoretically good time complexity. One limitation is that the proposed method requires a directed acyclic graph (DAG) representation of data. Hence, the data should be converted into DAG and also empirically the efficiency of the proposed SSM is not observed to surpass a baseline transformer.

**Strengths:**

1. **Improved applicability of SSMs.** This paper proposed a new SSM model that can incorporate the topology of data into SSMs. This expands the operating range of SSMs beyond 1D sequential data.
2. **Theoretically grounded modules.** The theorem suggests how to optimize the implementation and the normalization scheme is derived on theoretical grounds.

**Weaknesses:**

1. **Limited efficiency.** The main motivation of SSMs is the superior efficiency compared to Transformers enjoying the linear time complexity. However, this paper failed to prove its advantage in empirical efficiency over Transformers. The additional experiments posted by the authors show that the wall-clock time analysis failed to show the efficiency improvement of the proposed method. On average, the proposed method is 1.5 times slower than Graph-GPS.
2. **Limited topology/overclaim.** Unlike the authors’ claim, the proposed method was demonstrated on only three modalities: image, text, and graph. The proposed method is not general enough to handle arbitrary topology such as well-studied groups [1,2,3] referred to in the comments of **Reviewer yGvr**.

**Main reasons:**

This paper studies an interesting and emerging topic that aims to extend the operating range of SSMs with strong performance and potential efficiency. However, the authors failed to demonstrate the empirical gain in efficiency compared to Transformers. Considering the fact that the main motivation of SSMs is the superior efficiency over Transformer, the inferior wall-clock time remains a main concern that has not been resolved by the authors.

**Additional Comments On Reviewer Discussion:**

The reviewers raised several concerns. Especially, efficiency is a major concern and the authors failed to demonstrate an empirical gain in efficiency compared to Transformers. Considering the fact that the main advantage of SSMs is the superior efficiency over Transformers, the inferior efficiency remains a main concern that are not fully addressed by the authors.

---

### Decision · Program_Chairs · 2025-01-22

Reject